# Probabilistic neural operators for functional uncertainty quantification

**Christopher Bülte**                                                     *buelte@math.lmu.de*
*Ludwigs-Maximilians-Universität München*
*Munich Center for Machine Learning (MCML)*
*Munich, Germany*

**Philipp Scholl**                                                       *scholl@math.lmu.de*
*Ludwigs-Maximilians-Universität München*
*Munich Center for Machine Learning (MCML)*
*Munich, Germany*

**Gitta Kutyniok**                                                       *kutyniok@math.lmu.de*
*Ludwigs-Maximilians-Universität München*
*University of Tromsø*
*DLR-German Aerospace Center*
*Munich Center for Machine Learning (MCML)*
*Munich, Germany*

**Reviewed on OpenReview:** *https://openreview.net/forum?id=gangoPXSRw*

## Abstract

Neural operators aim to approximate the solution operator of a system of differential equations purely from data. They have shown immense success in modeling complex dynamical systems across various domains. However, the occurrence of uncertainties inherent in both model and data has so far rarely been taken into account—a critical limitation in complex, chaotic systems such as weather forecasting. In this paper, we introduce the probabilistic neural operator (PNO), a framework for learning probability distributions over the output function space of neural operators. PNO extends neural operators with generative modeling based on strictly proper scoring rules, integrating uncertainty information directly into the training process. We provide a theoretical justification for the approach and demonstrate improved performance in quantifying uncertainty across different domains and with respect to different baselines. Furthermore, PNO requires minimal adjustment to existing architectures, shows improved performance for most probabilistic prediction tasks, and leads to well-calibrated predictive distributions and adequate uncertainty representations even for long dynamical trajectories. Implementing our approach into large-scale models for physical applications can lead to improvements in corresponding uncertainty quantification and extreme event identification, ultimately leading to a deeper understanding of the prediction of such surrogate models.

## 1 Introduction

Many complex phenomena in the sciences are described via parametric partial differential equations (PDEs), making their study a crucial research topic for systems in biology, physics, or engineering sciences. Since traditional solvers, such as finite element methods are based on a space discretization there exists a tradeoff between solution accuracy and the computational complexity imposed by the discretization resolution. A recent breakthrough has been achieved by neural operators (Kovachki et al., 2023; Lu et al., 2021), which are a class of neural networks that have the capacity to approximate any continuous operator, including any

solution operator of a given family of parametric PDEs. Neural operators, and especially Fourier neural operators (Li et al., 2021), are now commonly used in practice and have been applied to problems such as weather forecasting (Kurth et al., 2023), fluid dynamics (Renn et al., 2023), seismic imaging (Li et al., 2023), carbon capture (Wen et al., 2023), or plasma modeling in nuclear fusion (Gopakumar et al., 2024). However, in application scenarios, the entire operator learning process is usually corrupted by different sources of noise. In addition, the model itself can be misspecified or—as in the case of dynamical systems—long-term predictions might be too chaotic to be predictable in a reliable manner. For these cases, it becomes crucial to have a precise understanding of the uncertainty of the corresponding model predictions.

A common approach to uncertainty quantification for standard neural networks is the Laplace approximation (Daxberger et al., 2021), which has been extended to Fourier neural operators by Weber et al. (2024) and Magnani et al. (2024). While this approach is efficient to implement and does not require additional training, it has notable limitations. Specifically, it does not incorporate uncertainty during network training, and its assumption of a Gaussian predictive distribution may be overly restrictive. Guo et al. (2023) introduce an information bottleneck structure that utilizes a decoder with a predictive Gaussian distribution. While innovative, this method shares the limitations of the Gaussian assumption.

Our work addresses these challenges by proposing a theoretically grounded approach to incorporate predictive uncertainty during training in the infinite-dimensional setting of neural operators. By leveraging strictly proper scoring rules, our method enables better-calibrated uncertainty estimates while supporting more flexible predictive distributions.

## 1.1 Our contributions

In this article, we introduce the *probabilistic neural operator (PNO)*, which extends the neural operator framework to a probabilistic setting, where the output is in the form of a (functional) probability distribution. Our novel approach is based on recent advances in training generative neural networks via proper scoring rules (Pacchiardi et al., 2024). PNO is designed to generate samples from an (empirical) probability distribution and is trained on an objective function that fits the predictive distribution to the corresponding observations, thereby integrating uncertainty directly into the training process. We build upon prior research by introducing a theoretical framework for proper scoring rules in separable Hilbert spaces. Specifically, we prove that the energy score is a strictly proper scoring rule in these spaces and, therefore, leads to desirable properties as a loss function. The resulting predictions of the network are well-calibrated and can be—due to their sample-based character—applied for different downstream tasks, including extreme event analysis and estimation of dependencies. We evaluate PNO on PDE datasets and a real-world data task, namely the prediction of surface temperature across Europe. PNO requires minimal adjustment to existing architectures and leads to well-calibrated predictive distributions and adequate uncertainty representations even for long dynamical trajectories. To summarize, we make the following key contributions:

- We introduce the *probabilistic neural operator (PNO)* as an extension of the neural operator to the probabilistic setting based on proper scoring rules.

- We prove that the *energy score* is a strictly proper scoring rule in separable Hilbert spaces and, therefore, a suitable loss function for the PNO.

- In extensive experiments including 2- and 3-dimensional PDEs, real-world weather data, and different model architectures we show that the PNO setup improves the calibration error while maintaining the accuracy of competing approaches.

## 1.2 Related work

In order to assess uncertainties in machine learning models, a variety of approaches are available; for a detailed overview compare Abdar et al. (2021). The most common way to perform uncertainty quantification is under the **Bayesian paradigm**. By utilizing Monte Carlo dropout during the inference phase of a neural network, Gal & Ghahramani (2016) show that the corresponding predictive distribution is a Bayesian approximation of a deep Gaussian process. While the approach requires no additional training, the approximation does not

hold for all architectures and no uncertainty information is used in the training process. Another method that has been popularized in deep learning recently is the Laplace approximation (Daxberger et al., 2021). It builds on the idea that, in a Bayesian setting, the posterior distribution over the network weights can be approximated by a Gaussian distribution, centered around a local maximum. The advantage is that this local maximum is available by regular training, while the covariance matrix of the Gaussian can be recovered by efficient approximations of the Hessian of the loss function (Botev et al., 2017).

A different class of approaches to uncertainty quantification is given by **ensemble methods**, where the main idea is to obtain an ensemble of predictions that differ slightly from each other and act as an empirical predictive distribution. A common way to utilize this for neural networks is to use so-called deep ensembles (Lakshminarayanan et al., 2017), where the ensemble is created by training several networks separately, with each optimization leading to a different local minimum. Although this has shown to perform well in practice, it is hindered by the computational demands of training multiple neural networks. Another method to obtain ensembles is by perturbing the input data before the forward pass through the model, which leads to an ensemble-based distribution in the output. This approach is used mainly in dynamical systems and weather forecasting (Bi et al., 2023; Bülte et al., 2025). However, this requires additional fine-tuning of the input perturbations and can lead to instabilities in the model predictions.

Finally, another way of uncertainty quantification is provided by **scoring rule**-based approaches. A scoring rule (Gneiting & Raftery, 2007) assigns a numerical score to a distribution-observation pair and can be minimized as a loss function, in order to fit a predictive distribution. Scoring rule approaches have been combined with nonparametric distributional regression (Walz et al., 2024), or neural networks (Rasp & Lerch, 2018), including multivariate cases (Pacchiardi et al., 2024; Chen et al., 2024). However, to the best of our knowledge, scoring rule approaches have not yet been transferred to the infinite-dimensional setting.

Since neural operators map to function spaces, the previously mentioned approaches need to be adjusted for **uncertainty quantification in infinite-dimensional spaces**. Weber et al. (2024) and Magnani et al. (2024) extend the Laplace approximation to Fourier neural operators, by performing the approximation in the last layers of the Fourier domain. While this approach is efficient to implement and requires no additional training, the assumption of a Gaussian can be restrictive, especially when it comes to calibration. Furthermore, the method is only applied post-hoc and no uncertainty information is used in the training process. Guo et al. (2023) propose an uncertainty quantification method that utilizes an information bottleneck and a Gaussian distribution to recover mean and standard deviations in the output space. However, this approach only leads to an estimate of the mean and standard deviation and does not allow for other measures of uncertainty. Ma et al. (2024) propose a quantile neural operator that is based on conformal prediction and provides a risk-controlling prediction set that includes the true solution with a user-specified probability. While this approach leads to a risk control for the error of the neural operator, it does not lead to a predictive distribution that can be used for assessing the full uncertainty in the prediction. In a different line of work, Ma et al. (2024) utilize the deep ensemble and Monte Carlo dropout methods as baselines, as they can be extended to the infinite-dimensional setting in a straightforward manner, leading to a functional ensemble prediction. However, the usability of the deep ensemble approach is limited by the computational overhead of training multiple models independently, especially if a large number of ensembles are required.

To the best of our knowledge, there are no other methods than those mentioned above that are specifically designed to combine uncertainty quantification with neural operators. Our work overcomes the previously discussed limitations, as it provides a predictive distribution that is directly incorporated into the training process by combining neural operators with scoring rule training.

### 1.3 Outline

This article is structured as follows: Section 2 provides preliminaries regarding the problem setting and neural operators, while Section 3 introduces neural network training via scoring rule minimization in separable Hilbert spaces. Section 4 is dedicated to the numerical experiments for several benchmark tasks, as well as a real-world application. The article finishes with the conclusion in Section 5.

## 2 Operator learning

Let $\mathcal{A} = \mathcal{A}(\mathcal{D}, \mathbb{R}^{d_a})$ and $\mathcal{U} = \mathcal{U}(\mathcal{D}; \mathbb{R}^{d_u})$ denote separable Banach spaces of functions over a bounded domain $\mathcal{D} \subset \mathbb{R}^d$ with values in $\mathbb{R}^{d_a}$ and $\mathbb{R}^{d_u}$, respectively. In this work, we consider parametric partial differential equations of the form:

$$\begin{aligned} \mathcal{L}_a[u](x) &= f(x), & x \in \mathcal{D}, \\ u(x) &= u_b(x), & x \in \partial D, \end{aligned} \tag{1}$$

for some $a \in \mathcal{A}, u \in \mathcal{U}, f \in \mathcal{U}$ and boundary condition $u_b \in \mathcal{U}$. We assume that the solution $u \in \mathcal{U}$ exists and is bounded for every $u_b$ and that $\mathcal{L}_a \in \mathcal{L}(\mathcal{U}, \mathcal{U})$ is a mapping from the parameter space $\mathcal{A}$ to the (possibly unbounded) space of linear operators mapping $\mathcal{U}$ to its dual. This gives rise to the nonlinear solution operator $G^\dagger : (a, u_b) \mapsto u$ mapping the parameters to the corresponding solution. Note that this includes initial condition as well as time-dependent problems, as $\mathcal{D}$ can be chosen as a spatio-temporal domain. We typically assume that we have observations $\{(a_j, u_j)\}_{j=1}^N$, where $a_j \sim \mu$ is generated from some probability measure $\mu$ supported on $\mathcal{A}$ and $u_j = \mathcal{G}^\dagger(a_j)$.

### 2.1 Neural operators

The aim of neural operators (Kovachki et al., 2023) is to generalize deep neural networks to the infinite-dimensional operator setting in order to learn an approximation of $\mathcal{G}^\dagger$ via a parametric map

$$\mathcal{G}_\theta : \mathcal{A} \to \mathcal{U}, \quad \theta \in \Theta, \tag{2}$$

with parameters from a finite-dimensional space $\Theta \subseteq \mathbb{R}^p$. By minimizing a suitable loss function one wants to choose $\theta^* \in \Theta$ such that $\mathcal{G}_{\theta^*} \approx \mathcal{G}^\dagger$. Following the framework of Li et al. (2020), neural operators are constructed using a sequence of layers consisting of learnable integral operators combined with pointwise non-linearities. For an input function $v_i \in \mathcal{U}(\mathcal{D}; \mathbb{R}^{d_v})$ at the $i^{th}$ layer, the map $G_\phi : \mathcal{U}(\mathcal{D}; \mathbb{R}^{d_v}) \to \mathcal{U}(\mathcal{D}; \mathbb{R}^{d_v}), v_i \mapsto v_{i+1}$ is computed as

$$G_\phi v_i(x) := \sigma\Big( \left( \mathcal{K}(a; \phi) v_i \right)(x) + W_{i,\phi} v_i(x) \Big), \quad \forall x \in \mathcal{D}.$$

Here, $W_{i,\phi} : \mathbb{R}^{d_v} \to \mathbb{R}^{d_v}$ is a linear transformation, $\sigma : \mathbb{R} \to \mathbb{R}$ denotes a pointwise nonlinearity, and $\mathcal{K} : \mathcal{A} \times \Theta_{\mathcal{K}} \to \mathcal{L}\left( \mathcal{U}(\mathcal{D}; \mathbb{R}^{d_v}), \mathcal{U}(\mathcal{D}; \mathbb{R}^{d_v}) \right)$ is a map to the bounded linear operators on $\mathcal{U}(\mathcal{D}; \mathbb{R}^{d_v})$, which is parameterized by $\phi \in \Theta_{\mathcal{K}} \subseteq \mathbb{R}^p$. Typically, $\mathcal{K}(a; \phi)$ is chosen as a kernel integral transformation that is parameterized by a neural network.

More specifically, in this article we mainly focus on the Fourier neural operator (FNO) (Li et al., 2021), which specifies the integral kernel as a convolution operator defined in Fourier space as

$$G_\phi v_i(x) = \sigma\Big( \mathcal{F}^{-1}(R_{i,\phi} \cdot \mathcal{F}(v_i))(x) + W_{i,\phi} v_i(x) \Big), \quad \forall x \in \mathcal{D},$$

where $\mathcal{F}$ is the Fourier transform of a function $f : \mathcal{D} \to \mathbb{R}^{d_v}$, $\mathcal{F}^{-1}$ its inverse, and $R_\phi = \mathcal{F}(\kappa_\phi)$, where $\kappa_\phi : \mathcal{D} \to \mathbb{R}^{d_v}$ is a periodic function parameterized by neural network parameters $\phi \in \Theta_{\mathcal{K}}$. This means that in each layer $i$, $R_{i,\phi}$ is directly parameterized in the Fourier domain by a neural network with parameters $\phi \in \Theta_{\mathcal{K}}$. Typically, several of these layers are combined together with additional pointwise lifting and projection operators (Li et al., 2021). Several other variants and extensions of Fourier neural operators have been proposed, such as U-shaped neural operators (Rahman et al., 2023), spherical Fourier neural operators (Bonev et al., 2023) or Wavelet neural operators (Tripura & Chakraborty, 2023).

## 3 A probabilistic neural operator

In the neural operator setting introduced above, the data pairs can be corrupted by noise and in time-dependent PDEs long prediction horizons might lead to unstable predictions. Moving to a probabilistic setting, instead of mapping directly to the solution, we require the neural operator to output a probability distribution over the function space $\mathcal{U}$. Assume that we have observational data $\{(a_j, u_j)\}_{j=1}^N$ with prior

distribution $a_j \sim P_{\mathcal{A}}$ and conditional distribution $u_j \sim P_{\mathcal{U}}^*(\,\cdot\mid a_j\,)$ and let $\mathcal{P}_{\mathcal{U}}$ denote a set of probability measures over $\mathcal{U}$. We consider a probabilistic neural operator to be a map

$$\mathcal{G}_\theta : \mathcal{A} \times \mathcal{Z} \to \mathcal{U}, \quad \theta \in \Theta,$$

that aims to learn a distribution $P_\theta(\,\cdot\mid a_j\,) := \mathcal{G}_\theta(a_j, \cdot) \in \mathcal{P}_{\mathcal{U}}$ parameterized by $\theta \in \Theta$ that fulfills $P_\theta(\,\cdot\mid a_j\,) \approx P_{\mathcal{U}}^*(\,\cdot\mid a_j\,)$. Samples from the conditional distribution are generated via $\mathcal{G}_\theta(a, z)$, where $z \sim \mu_{\mathcal{Z}}$ and $\mu_{\mathcal{Z}}$ is a probability distribution over some space $\mathcal{Z}$, usually chosen as a simple tractable distribution such as a Gaussian. For a fixed initial condition $a$, we denote the approximate (empirical) conditional distribution parameterized by the neural network as $P_\theta^M := \{\mathcal{G}_\theta(a, z_m)\}_{m=1}^M$ with $z_m \sim \mu_{\mathcal{Z}}$, $m = 1, ..., M$, where we drop the dependence on the initial condition $a$ from the notation. In this article, we restrict ourselves to data from separable Hilbert spaces, which include most solution spaces of PDEs, such as the Sobolev spaces $H^k, k \in \mathbb{N}$.

## 3.1 Proper scoring rules for neural operators

Our methodology builds around the theory of proper scoring rules and its extensions to separable Hilbert spaces (Steinwart & Ziegel, 2021; Ziegel et al., 2024). Let $(\mathcal{X}, \mathcal{A})$ be a measurable space and let $\mathcal{M}_1(\mathcal{X})$ denote the class of all probability measures on $\mathcal{X}$. For $\mathcal{P} \subseteq \mathcal{M}_1(\mathcal{X})$, a *scoring rule* is a function $S : \mathcal{P} \times \mathcal{X} \to [-\infty, \infty]$ such that the integral $\int S(P, x) dQ(x)$ exists for all $P, Q \in \mathcal{P}$. Define the so-called *expected score* as $S(Q, P) := \int S(Q, x)\,dP(x) = \mathbb{E}_{X \sim P}[S(Q, X)]$. Then $S$ is called *proper* relative to the class $\mathcal{P}$ if

$$S(P, P) \leq S(Q, P), \quad \text{for all } P, Q \in \mathcal{P}, \tag{3}$$

and it is called *strictly proper* if equality in (3) implies $P = Q$ (Gneiting & Raftery, 2007). In other words, for a strictly proper scoring rule, $P$ and $Q$ coincide if and only if $Q$ is the unique minimizer of the expected score. This allows for efficient and simple training of neural networks via strictly proper scoring rules, as shown later.

In this article, we focus on the so-called *energy score* (Gneiting & Raftery, 2007), which is a commonly used scoring rule for the multivariate finite-dimensional setting that has been successfully used in combination with neural networks (Pacchiardi et al., 2024; Chen et al., 2024). In the following theorem we prove that the energy score can be extended to a strictly proper scoring rule over separable Hilbert spaces.

**Theorem 3.1** (Energy score). *Let $\mathcal{H}$ denote a separable Hilbert space and $x \in \mathcal{H}$. The energy score* $\mathrm{ES} : \mathcal{M}_1^E(\mathcal{H}) \times \mathcal{H} \to \mathbb{R}$ *defined as*

$$\mathrm{ES}(P, x) := \mathbb{E}_P[\|X - x\|_{\mathcal{H}}] - \frac{1}{2}\mathbb{E}_P[\|X - X'\|_{\mathcal{H}}], \tag{4}$$

*with* $X, X' \overset{i.i.d}{\sim} P \in \mathcal{M}_1^E(\mathcal{H}) := \left\{ P \in \mathcal{M}_1(\mathcal{H}) \mid \int_{\mathcal{H}} \|x\|_{\mathcal{H}} dP(x) < \infty \right\}$ *is strictly proper relative to the class* $\mathcal{M}_1^E(\mathcal{H})$ .

*Proof.* As a first step, we show that a separable Hilbert space induces a positive definite kernel that is characteristic. A measurable and bounded kernel $k$ is called *characteristic* if the kernel embedding defined by $\Phi(P) := \int k(\cdot, \omega) dP(\omega)$, with $P \in \mathcal{M}_1^k(\mathcal{H}) := \left\{ P \in \mathcal{M}_1(\mathcal{H}) \mid \int_{\mathcal{H}} \sqrt{k(x, x)} dP(x) < \infty \right\}$, is injective (Steinwart & Ziegel, 2021).

Let $(\mathcal{X}, d)$ be a metric space and define $\mathcal{M}_1^d(\mathcal{X}) := \left\{ P \in \mathcal{M}_1(\mathcal{X}) \mid \exists o \in \mathcal{X} \text{ such that } \int d(o, x)\,dP(x) < \infty \right\}$. It is said to be of *negative type* if $d$ is negative definite, i.e. for all $n \in \mathbb{N}$, $x_1, ..., x_n \in \mathcal{X}$ and $\alpha_1, ..., \alpha_n \in \mathbb{R}$ with $\sum_{i=1}^n \alpha_i = 0$ it holds that $\sum_{i,j=1}^n \alpha_i \alpha_j d(x_i, x_j) \leq 0$ (compare Berg et al., 1984, Definition 3.1.1). Furthermore, let $P_1, P_2 \in \mathcal{M}_1^d(\mathcal{X})$. Then, following Lyons (2013), if $\mathcal{X}$ has negative type, it holds that

$$\int d(x_1, x_2)\,d(P_1 - P_2)^2(x_1, x_2) \leq 0. \tag{5}$$

Furthermore, the space $(\mathcal{X}, d)$ is said to be of *strong negative type* if it is of negative type and equality in (5) holds if and only if $P_1 = P_2$, which is the case for separable Hilbert spaces (Lyons, 2013, Theorem 3.25).

Furthermore, a metric space $(\mathcal{X}, d)$ of negative type induces a positive definite kernel $k : \mathcal{X} \times \mathcal{X} \to \mathbb{R}$ given as $k(x, y) = d(x, z_0) + d(y, z_0) - d(x, y)$ for some fixed $z_0 \in \mathcal{X}$. Finally, Sejdinovic et al. (2013, Proposition 29) state that for a metric space of strong negative type the corresponding induced kernel $k$ is characteristic.

Next, we cite Steinwart & Ziegel (2021, Theorem 1.1), who prove that the *kernel score* $S_k : \mathcal{M}_1^k \times \mathcal{X} \to \mathbb{R}$, defined as $S_k(P, x) := \frac{1}{2}\mathbb{E}_P[k(X, X')] - \mathbb{E}_P[k(X, x)] + \frac{1}{2}k(x, x)$, where $x \in \mathcal{X}$ and $X, X' \overset{i.i.d}{\sim} P \in \mathcal{M}_1^k$, is a strictly proper scoring rule if $k$ is a characteristic kernel. Therefore, it only remains to show that the energy score corresponds to the kernel score $S_{k_d}$ given by the induced kernel $k_d := d(x, z_0) + d(y, z_0) - d(x, y)$ in the Hilbert space $\mathcal{H}$ with metric $d(x, x') := \|x - x'\|_{\mathcal{H}}$ and for some fixed $z_0 \in \mathcal{H}$. We obtain

$$
\begin{aligned}
S_{k_d}(P, x) &= -\mathbb{E}_P[k_d(X, x)] + \frac{1}{2}\mathbb{E}_P[k_d(X, X')] + \frac{1}{2}k_d(x, x) \\
&= -\mathbb{E}_P[d(X, z_0) + d(x, z_0) - d(X, x)] + \frac{1}{2}\mathbb{E}_P[d(X, z_0) + d(X', z_0) - d(X, X')] \\
&\quad + \frac{1}{2}(d(x, z_0) + d(x, z_0) - d(x, x)) \\
&= \mathbb{E}_P[d(X, x)] - \frac{1}{2}\mathbb{E}_P[d(X, X')] - \frac{1}{2}\mathbb{E}_P[d(X, z_0)] + \frac{1}{2}\mathbb{E}_P[d(X', z_0)] - \mathbb{E}_P[d(x, z_0)] + d(x, z_0) \\
&= \mathbb{E}_P[d(X, x)] - \frac{1}{2}\mathbb{E}_P[d(X, X')] = \mathrm{ES}(P, x).
\end{aligned}
$$

Since $\mathcal{H}$ is of strong negative type, $k_d$ is characteristic and, thus, by Steinwart & Ziegel (2021), the energy score is strictly proper relative to the class $\mathcal{M}_1^{k_d}(\mathcal{H})$. □

## 3.2 Training neural operators via the energy score

After introducing the underlying concepts and showing that the energy score is strictly proper in separable Hilbert spaces, we now introduce the corresponding training regime for neural networks. This follows the general framework of Pacchiardi et al. (2024), adapted to operator learning. Given a strictly proper scoring rule, the minimization objective is given as

$$
\underset{\theta}{\mathrm{argmin}} \; \mathbb{E}_{a \sim P_{\mathcal{A}}} \mathbb{E}_{u \sim P_{\mathcal{U}}^*(\cdot | a)} S(P_\theta(\cdot \mid a), u) \tag{6}
$$

and leads to $P_\theta^*(\cdot \mid a) = P_{\mathcal{U}}^*(\cdot \mid a)$ almost everywhere (Pacchiardi et al., 2024). In the finite data setting, the objective can be approximated with a Monte-Carlo estimator. While closed-form expressions of $S$ are not always available, the energy score has a representation that admits an unbiased estimator, which requires the output from our neural network to consist of multiple samples of the predictive distribution, i.e., $M > 1$.[1] The minimization objective for the neural network with the energy score is then given by

$$
\underset{\theta}{\mathrm{argmin}} \; \frac{1}{N} \sum_{i=1}^{N} \left( \frac{1}{M} \sum_{j=1}^{M} \|u_i^j - u_i\|_{\mathcal{H}} - \frac{1}{2M(M-1)} \sum_{\substack{j,h=1 \\ j \neq h}}^{M} \|u_i^j - u_i^h\|_{\mathcal{H}} \right). \tag{7}
$$

Different methods are available to specify a neural network to output a conditional (empirical) distribution. Here, we focus on two main approaches. One way to obtain samples is to utilize the reparametrization trick (Kingma et al., 2015), which learns a pointwise normal distribution in the last layer of the network, by including two projection layers corresponding to the mean and standard deviation of a normal distribution. Although this has a computational advantage as only one forward pass is required, the predictive distribution is fixed to a Gaussian, which might not be expressive enough for every task. A different way to obtain samples is via stochastic dropout (Gal & Ghahramani, 2016). By adding Bernoulli or Normal random samples to specific weights of the network, a forward pass becomes stochastic and, therefore, leads to a predictive distribution. While this approach has higher computational demands due to the need to perform multiple

---

[1]Note that this setting still includes the classical neural operator as the special case, where the predictive distribution is a Dirac measure. The objective function then becomes the $\|\cdot\|_{\mathcal{H}}$ loss.

forward passes, the partial stochasticity of the network leads to a higher expressivity with regard to the conditional distribution (Sharma et al., 2023). We add additional dropout in the Fourier domain, setting random frequency parts as zero with the pre-specified dropout probability, making use of the global structure of the Fourier domain. Since both approaches are quite different and come with different strengths and weaknesses, we included them both in the experiments. For the remainder of this article, we will refer to the methods as PNO$_D$ and PNO$_R$ for the dropout variant and the parametrization variant, respectively.

## 4 Numerical experiments

This section describes our experimental setup. In Subsection 4.1 we introduce the baseline uncertainty quantification methods for comparisons, while Subsection 4.2 includes the evaluation metrics. In the remaining subsections we present the different experiments. All experiments are evaluated on a previously unseen test set with predictions aggregated over ten training runs with different random seeds. The different datasets as well as the corresponding grids are normalized before they are used as input to the network. As architectures, for our experiments, we focus on the Fourier Neural Operator (FNO) (Li et al., 2021) and provide additional results for the U-shaped Neural Operator (UNO) (Rahman et al., 2023) architecture in Appendix B. For the spherical shallow water equation we employ the spherical FNO (Bonev et al., 2023). The training process and the corresponding hyperparameters are explained in detail in Appendix A. Although all uncertainty quantification methods use the same underlying architecture in the experiments, the dropout rate has been tuned separately, as it has a significant impact on the shape of the predictive distribution for the different methods. While we keep the dropout rate in the Fourier layers and the standard layers equal during the experiments in section 4, we assess the effect of both changes independently in Appendix C. In order to evaluate our approach beyond the pure performance, we include an additional runtime analysis of the different methods in Appendix D. In Appendix F we assess the influence of the number of samples $M$ during training for PNO. Finally, in Appendix G we provide a more detailed comparison of PNO$_D$ and PNO$_R$, highlighting differences and providing guidelines on which approach to choose.

### 4.1 Baseline models

As a basic comparison and a sanity check, we use a deterministic neural operator (DET), where uncertainty is not directly accounted for. Here, the predictive distribution is a degenerate Dirac distribution with

$$P_\theta^M = \{\mathcal{G}_\theta^*(a)\}_{m=1}^M, \tag{8}$$

i.e. the predictive are all a constant value independent of $m$. In addition, we compare our method to different uncertainty quantification methods, adapted to the task of operator learning. Gal & Ghahramani (2016) show that a neural network with dropout before each layer is mathematically equivalent to a variational approximation of a Gaussian process. This leads to a simple and efficient way of creating a predictive distribution, known as MCDropout (MCD), which has been extended to neural operators by Ma et al. (2024). In our setting, the predictive distribution for a fixed input $a$ is given as

$$P_\theta^M = \{\mathcal{G}_\theta^*(a, \boldsymbol{z}_m)\}_{m=1}^M, \tag{9}$$

where $\boldsymbol{z}_m$ is the random dropout mask and $\mathcal{G}_\theta^*$ denotes the trained neural operator. As opposed to the PNO, the MCDropout model is trained on the $L^p$ loss and no uncertainty information is used in the training process.

Weber et al. (2024); Magnani et al. (2024) propose to utilize the Laplace approximation (LA) for FNOs, which is based on building a second-order approximation of the weights around the maximum a posteriori (MAP) estimate $\theta_{\text{MAP}}$. By assuming a Gaussian weight prior, the weight-space uncertainty of the LA is given by

$$p(\theta, \mathcal{C}) \approx \mathcal{N}(\theta; \theta_{\text{MAP}}, \Sigma), \quad \Sigma \coloneqq -(\nabla_\theta^2 \mathcal{L}(\mathcal{C}; \theta)|_{\theta_{\text{MAP}}})^{-1}, \tag{10}$$

where $\mathcal{C} = \{(a_n, u_n)\}_{n=1}^N$. By sampling from this Gaussian we obtain the predictive distribution $P_\theta^M$. We utilize a last-layer Laplace approximation, which leads to an analytically available Gaussian predictive distribution in function space. Since the Laplace approximation is built around the MAP estimate, it also does not utilize uncertainty information during the training process.

## 4.2 Evaluation

To thoroughly assess the performance of the prediction methods, we use a range of evaluation metrics, each focusing on a different aspect of the probabilistic prediction. Denote by $\mathcal{D}$ the spatio-temporal domain of the governing equation, by $P_\theta^M = \{u^m\}_{m=1}^M$ the predictive distribution, and by $u$ the corresponding true observation. Furthermore, let $\bar{u}_M := \frac{1}{M} \sum_{m=1}^M u^m$ denote the mean prediction, $\hat{\sigma}^2 := \frac{1}{M-1} \sum_{m=1}^M (u^m - \bar{u}_M)^2$ denote the estimate of the (pointwise) variance and $\hat{q}_\theta^\alpha$ denote the empirical (pointwise) quantiles of $P_\theta^M$ at the level $\alpha$. We utilize the following evaluation metrics:

$$\mathcal{L}_{L^2}(P_\theta^M, u) := \|\bar{u}_M - u\|_{L^2}, \tag{11}$$

$$\mathrm{ES}(P_\theta^M, u) := \frac{1}{M} \sum_{m=1}^M \|\hat{u}^m - u\|_{L^2} - \frac{1}{2M(M-1)} \sum_{\substack{m,h=1 \\ m \neq h}}^M \|\hat{u}^m - \hat{u}^h\|_{L^2}, \tag{12}$$

$$\mathrm{CRPS}(P_\theta^M, u) := \frac{1}{|\mathcal{D}|} \sum_{(x,t) \in \mathcal{D}} \left( \int_0^1 \mathrm{QS}_\alpha\left(\hat{q}_\theta^\alpha(x,t), y\right) d\alpha \right), \tag{13}$$

$$\mathrm{NLL}(P_\theta^M, u) := \frac{1}{2|\mathcal{D}|} \sum_{(x,t) \in \mathcal{D}} \log\left(2\pi\hat{\sigma}^2(x,t)\right) + \frac{(\bar{u}_M(x,t) - u(x,t))^2}{\hat{\sigma}^2(x,t)}, \tag{14}$$

$$\mathcal{C}_\alpha(P_\theta^M, u) := \frac{1}{|\mathcal{D}|} \sum_{(x,t) \in \mathcal{D}} \mathbb{1}\left\{ u(x,t) \in [\hat{q}_\theta^{\alpha/2}(x,t), \hat{q}_\theta^{1-\alpha/2}(x,t)] \right\}, \tag{15}$$

$$\left|\mathcal{C}_\alpha(P_\theta^M, u)\right| := \frac{1}{|\mathcal{D}|} \sum_{(x,t) \in \mathcal{D}} \left| \hat{q}_\theta^{1-\alpha/2}(x,t) - \hat{q}_\theta^{\alpha/2}(x,t) \right|. \tag{16}$$

The $\mathcal{L}_{L^2}$ loss evaluates the match between the mean prediction $\bar{u}_M$ and the observation, while the energy score (ES) evaluates the match for the predictive distribution as a whole. The continuous ranked probability score (CRPS) (Gneiting & Raftery, 2007) evaluates the predictive distribution at a pointwise level and can be represented as an integral of the quantile score $QS_\alpha(q_\alpha, y) := 2(\alpha - \mathbb{1}\{y < q_\alpha\})(y - q_\alpha)$ over all quantile levels $\alpha \in (0, 1)$. The CRPS therefore assesses whether the predicted uncertainty fits the observations at all quantile levels at each gridpoint $(x, t)$, i.e., whether the predictive distribution is well-calibrated. Furthermore, we analyze the negative log-likelihood (NLL) of a predictive pointwise Gaussian distribution $\mathcal{N}(\bar{u}_M(x,t), \hat{\sigma}^2(x,t))$. Although not all methods produce Gaussian output, this criterion is commonly used and describes the fit of the predictive distribution in terms of the first two estimated moments. The NLL is also a strictly proper scoring rule (Gneiting & Raftery, 2007). Finally, we report the coverage $\mathcal{C}_\alpha$ of the predictive distribution for selected $\alpha$-quantile levels, as well as the width of the corresponding prediction interval $|\mathcal{C}_\alpha|$. Following the notions of calibration and sharpness (Gneiting et al., 2007), in an ideal setting, an estimator should obtain $\mathcal{C}_\alpha \approx 1 - \alpha$, while simultaneously predicting the interval $|\mathcal{C}_\alpha|$ as small as possible. To make the baselines comparable, $M = 100$ predictive samples are drawn from the respective predictive distributions at test time, and all metrics are averaged over the test dataset. For the deterministic model, the energy score reduces to the $\mathcal{L}_{L^2}$ loss and the CRPS reduces to the pointwise mean absolute error, while the negative log-likelihood and the coverage are not applicable.

## 4.3 Darcy flow

First, we consider a steady-state solution of the two-dimensional Darcy flow over the unit square, which is defined by a second-order linear elliptic equation of the form

$$-\nabla(a(x)\nabla u(x)) = f(x), \qquad x \in (0,1)^2,$$
$$u(x) = 0, \qquad x \in \partial(0,1)^2.$$

Here, $a \in L^\infty((0,1)^2; \mathbb{R}_+)$ is the diffusion coefficient, $u \in H_0^1((0,1)^2, \mathbb{R})$ is the velocity function, and $f \in L^2((0,1)^2; \mathbb{R})$ is the forcing function. The aim is to learn the solution operator $G^* : L^\infty((0,1)^2; \mathbb{R}_+) \to H_0^1((0,1)^2; \mathbb{R})$, $a \mapsto u$. The PDE is usually used for modeling fluid flow in a porous medium. We use data

| Model | Method | $\mathcal{L}_{L^2}$ | ES | CRPS | NLL | $\mathcal{C}_{0.05}$ | $|\mathcal{C}_{0.05}|$ |
|-------|--------|------|----|------|-----|------|-------|
| **FNO** | DET | 0.0972 ($\pm$ 0.0076) | 0.0972 ($\pm$ 0.0076) | 0.0783 ($\pm$ 0.0062) | - | - | - |
| | PNO$_D$ | **0.0772** ($\pm$0.0015) | **0.0569** ($\pm$0.0009) | **0.0462** ($\pm$0.0007) | **-1.3444** ($\pm$0.0355) | **0.9813** ($\pm$0.0032) | 0.3682 ($\pm$0.0185) |
| | PNO$_R$ | 0.1031 ($\pm$0.0069) | 0.0817 ($\pm$0.0056) | 0.0707 ($\pm$0.0040) | 685.15 ($\pm$913.21) | 0.4702 ($\pm$0.0559) | 0.1249 ($\pm$0.0191) |
| | MCD | 0.1002 ($\pm$0.0088) | 0.0830 ($\pm$0.0080) | 0.0672 ($\pm$0.0066) | 1.5973 ($\pm$0.8223) | 0.5187 ($\pm$0.0608) | 0.1203 ($\pm$0.0031) |
| | LA | 0.0985 ($\pm$0.0074) | 0.0767 ($\pm$0.0040) | 0.0621 ($\pm$0.0033) | -0.8851 ($\pm$0.0531) | **0.9762** ($\pm$0.0085) | 0.5713 ($\pm$0.0076) |

Table 1: Results for the Darcy Flow equation using the FNO architecture with the different baselines. The best model is highlighted in bold for all metrics, except for the interval width, as it is only comparable for models with the same coverage. The standard deviation across the different experiment runs is given in the brackets.

from Takamoto et al. (2022), with the forcing function fixed as $f(x) = 1$, which corresponds to the setting in Li et al. (2021). The provided data consists of 10000 samples with a resolution of $128 \times 128$. We train the networks on a spatial resolution of $64 \times 64$ and evaluate on $128 \times 128$. The results are presented in Table 1. The PNO$_D$ obtains the best scores across all metrics when using the FNO architecture. Specifically, it is worth noting that it obtains a lower $L^2$ loss, although the deterministic model is directly trained on that metric, indicating that learning the whole predictive distribution can also improve the mean prediction. While both the PNO$_D$ and the Laplace approach achieve the coverage goal, the prediction interval of the PNO$_D$ is significantly narrower despite the coverage of PNO$_D$ being better as well. A visualization of the different predictions can be found in Appendix H.

### 4.4 Kuramoto-Sivashinsky equation

| Model | Method | $\mathcal{L}_{L^2}$ | ES | CRPS | NLL | $\mathcal{C}_{0.05}$ | $|\mathcal{C}_{0.05}|$ |
|-------|--------|------|----|------|-----|------|-------|
| **FNO** | DET | 0.9028 ($\pm$ 0.0086) | 0.9028 ($\pm$ 0.0086) | 0.7100 ($\pm$ 0.0106) | - | - | - |
| | PNO$_D$ | 0.8793 ($\pm$0.0072) | 0.6195 ($\pm$0.0049) | 0.5496 ($\pm$0.0097) | 3.0389 ($\pm$0.5872) | 0.7640 ($\pm$0.0191) | 3.0852 ($\pm$0.0422) |
| | PNO$_R$ | 0.8640 ($\pm$0.0038) | **0.6081** ($\pm$0.0027) | **0.4743** ($\pm$0.0037) | **1.2268** ($\pm$0.0083) | **0.9401** ($\pm$0.0030) | 3.2594 ($\pm$0.0294) |
| | MCD | **0.8635** ($\pm$0.0048) | 0.7541 ($\pm$0.0049) | 0.5974 ($\pm$0.0058) | 107.10 ($\pm$22.941) | 0.3600 ($\pm$0.0082) | 0.6046 ($\pm$0.0164) |
| | LA | 0.8772 ($\pm$0.0055) | 0.7332 ($\pm$0.0103) | 0.6081 ($\pm$0.0077) | 43.9302 ($\pm$10.8764) | 0.3955 ($\pm$0.0189) | 0.8175 ($\pm$0.0669) |

Table 2: Results for the Kuramoto-Sivashinsky equation. The best model is highlighted in bold and the standard deviation is given in the brackets.

The Kuramoto-Sivashinsky (KS-)equation is a fourth-order parabolic PDE with the following form in one spatial dimension:

$$\partial_t u(x,t) + u\partial_x u(x,t) + \partial_x^2 u(x,t) + \partial_x^4 u(x,t) = 0, \qquad x \in \mathcal{D}, t \in (0,T]$$
$$u(x,0) = u_0(x), \qquad x \in \mathcal{D}$$

where $\mathcal{D} \subseteq \mathbb{R}$, $u \in C((0,T]; H^4_{\mathrm{per}}(\mathcal{D}; \mathbb{R}))$, and $u_0 \in L^2_{\mathrm{per}}(\mathcal{D}; \mathbb{R})^2$. Furthermore, $u_0$ is $L$-periodic, where $L$ is the size of the specific domain. The PDE exhibits strong temporal chaotic behavior and is used for modeling

---

$^2 H^k_{\mathrm{per}}(\mathcal{D}; \mathbb{R}), L^2_{\mathrm{per}}(\mathcal{D}; \mathbb{R})$ denote the periodic Sobolev and $L^2$ spaces, respectively.

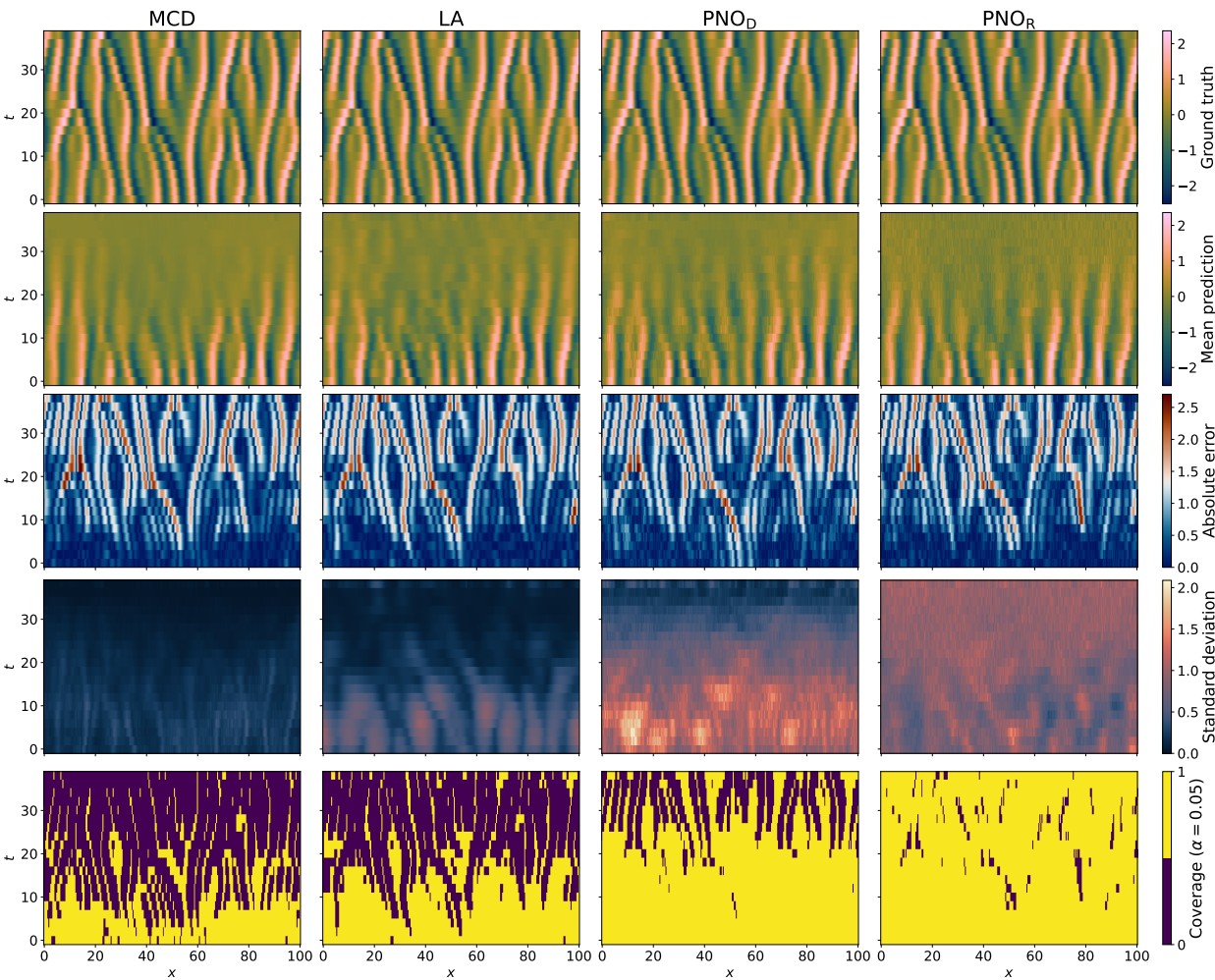

Figure 1: The figure shows the ground truth, mean prediction, absolute error, standard deviation, and coverage for the different methods on a sample of the Kuramoto-Sivashinsky equation.

instabilities in a laminar flame front, reaction-diffusion systems, or other chemical reactions. We simulate the KS-equation from random uniform noise $\mathcal{U}(-1, 1)$ on a periodic domain $\mathcal{D} = [0, 100]$ using the py-pde package (Zwicker, 2020). We generate 10000 samples with a resolution of $256 \times 300$ and train a 2D (spatio-temporal) FNO, which takes the first twenty timesteps as input and solves the equation for the next twenty timesteps with step size $\Delta t = 2$. The results for the KS-equation can be found in Table 2. Here, the MCDropout approach leads to the best $L^2$ loss, while the $PNO_R$ leads to the best distributional metrics. Although no method achieves a coverage of 95%, the $PNO_R$ is closest. A corresponding visualization can be found in Figure 1.

## 4.5 Spherical shallow water equation

Bonev et al. (2023) have extended the FNO to convolutions on the sphere using spherical harmonic basis functions, leading to the so-called Spherical Fourier Neural Operator (SFNO). Following their experiments, we test our method on the shallow water equation on the rotating sphere. The shallow water equations (SWE) are a set of hyperbolic partial differential equations, derived from the general Navier-Stokes equations

and present a framework for modeling free-surface flow problems. In the spherical setting, these are given as

$$\partial_t h(x,t) + \nabla h(x,t)\boldsymbol{u} = 0, \qquad x \in S^2, t \in (0,T]$$
$$\partial_t h(x,t)\boldsymbol{u} + \nabla\left((\boldsymbol{u} \odot \boldsymbol{u})h(x,t)\right) = S, \qquad x \in S^2, t \in (0,T],$$

with the vector of directional velocities $\boldsymbol{u} = (u,v)^\top$. Here, $h \in C((0,T]; H_0^1(S^2; \mathbb{R}))$ is the geopotential of the water and $S$ contains flux terms, such as the Coriolis force. Similarly to Bonev et al. (2023), we simulate 5000 training samples with resolution $128 \times 256$ with PDE parameters that model the Earth and an additional 500 evaluation samples. Gaussian random fields are used as initial conditions with a burn-in period of three hours for the numerical solver. The model is trained and evaluated by using the corresponding losses in a weighted version, adjusted to the latitude weighting of the sphere. Furthermore, Bonev et al. (2023) propose a fine-tuning step, where the model is trained with an additional autoregressive step, which leads to more stable results for long rollout times, which we additionally include in our experiments. For evaluation, we report both training versions with one-hour and ten-hour prediction horizons each. The results for the two-step autoregressive training can be found in Table 3 and the results for the single-step training in Table 12 in Appendix I. For the one hour prediction, the PNO$_\text{D}$ obtains the best metrics across all models, and especially a good coverage. For the ten-hour prediction task, the deterministic model performs best, although the PNO$_\text{R}$ and PNO$_\text{D}$ perform quite similar. It is notable that the PNO$_\text{R}$ leads to very large values for the negative log-likelihood, due to the predicted variance collapsing zero, which could indicate an optimization failure. However, the method is still competitive in the remaining metrics. This might be due to overconfident pointwise predictions, which lead to a small variance and, therefore, to a large NLL. This behavior is also visible in the visualization in Figure 2. The MCD approach does not lead to stable predictions for the ten-hour prediction task, as is reflected in all metrics. However, for the single-step training, the performance is much better (compare Table 12), indicating that the method cannot be combined with the autoregressive training approach.

| $\Delta t$ | Method | $\mathcal{L}_{L^2}$ | ES | CRPS | NLL | $\mathcal{C}_{0.05}$ | $\|\mathcal{C}_{0.05}\|$ |
|---|---|---|---|---|---|---|---|
| **1h** | DET | 0.3150 ($\pm$ 0.0129) | 0.3150 ($\pm$ 0.0129) | 0.1244 ($\pm$ 0.0046) | - | - | - |
| | PNO$_\text{D}$ | **0.2970** ($\pm$0.0232) | **0.2216** ($\pm$0.0142) | **0.0903** ($\pm$0.0056) | **-0.4561** ($\pm$0.0447) | **0.9828** ($\pm$0.0031) | 0.8959 ($\pm$0.0440) |
| | PNO$_\text{R}$ | 0.3871 ($\pm$0.0126) | 0.2740 ($\pm$0.0088) | 0.1561 ($\pm$0.0050) | $2.3\times10^{10}$ ($\pm1.6\times10^9$) | 0.0274 ($\pm$0.0206) | 0.0844 ($\pm$0.0551) |
| | MCD | 0.3022 ($\pm$0.0204) | 0.2368 ($\pm$0.0196) | 0.0944 ($\pm$0.0066) | 0.8878 ($\pm$0.2608) | 0.6023 ($\pm$0.0250) | 0.2550 ($\pm$0.0072) |
| | LA | 0.4262 ($\pm$0.0522) | 0.3159 ($\pm$0.0299) | 0.1400 ($\pm$0.0120) | 0.2374 ($\pm$0.2229) | 0.9096 ($\pm$0.0762) | 1.0866 ($\pm$0.3472) |
| **10h** | DET | **0.9215** ($\pm$ 0.0678) | **0.9215** ($\pm$ 0.0678) | **0.3933** ($\pm$ 0.0254) | - | - | - |
| | PNO$_\text{D}$ | 1.2421 ($\pm$0.0650) | 0.9915 ($\pm$0.0535) | 0.4330 ($\pm$0.0215) | **5.4385** ($\pm$0.5448) | 0.4935 ($\pm$0.0200) | 0.9210 ($\pm$0.0793) |
| | PNO$_\text{R}$ | 1.1803 ($\pm$0.0345) | 0.9535 ($\pm$0.0413) | 0.4995 ($\pm$0.0130) | $2.1\times10^{11}$ ($\pm1.4\times10^{10}$) | 0.0212 ($\pm$0.0200) | 0.0850 ($\pm$0.0546) |
| | MCD | $9.1\times10^5$ ($\pm2.7\times10^6$) | $7.5\times10^5$ ($\pm2.2\times10^6$) | $2.8\times10^5$ ($\pm8.3\times10^5$) | 22.292 ($\pm$19.623) | 0.3530 ($\pm$0.1364) | $6.7\times10^5$ ($\pm2.0\times10^6$) |
| | LA | 1.5405 ($\pm$0.3697) | 1.2612 ($\pm$0.3338) | 0.5299 ($\pm$0.1336) | 6.4958 ($\pm$5.5154) | **0.5759** ($\pm$0.1480) | 1.1925 ($\pm$0.4394) |

Table 3: Results for the spherical shallow water equation with two-step autoregressive training. The best model is highlighted in bold and the standard deviation is given in the brackets.

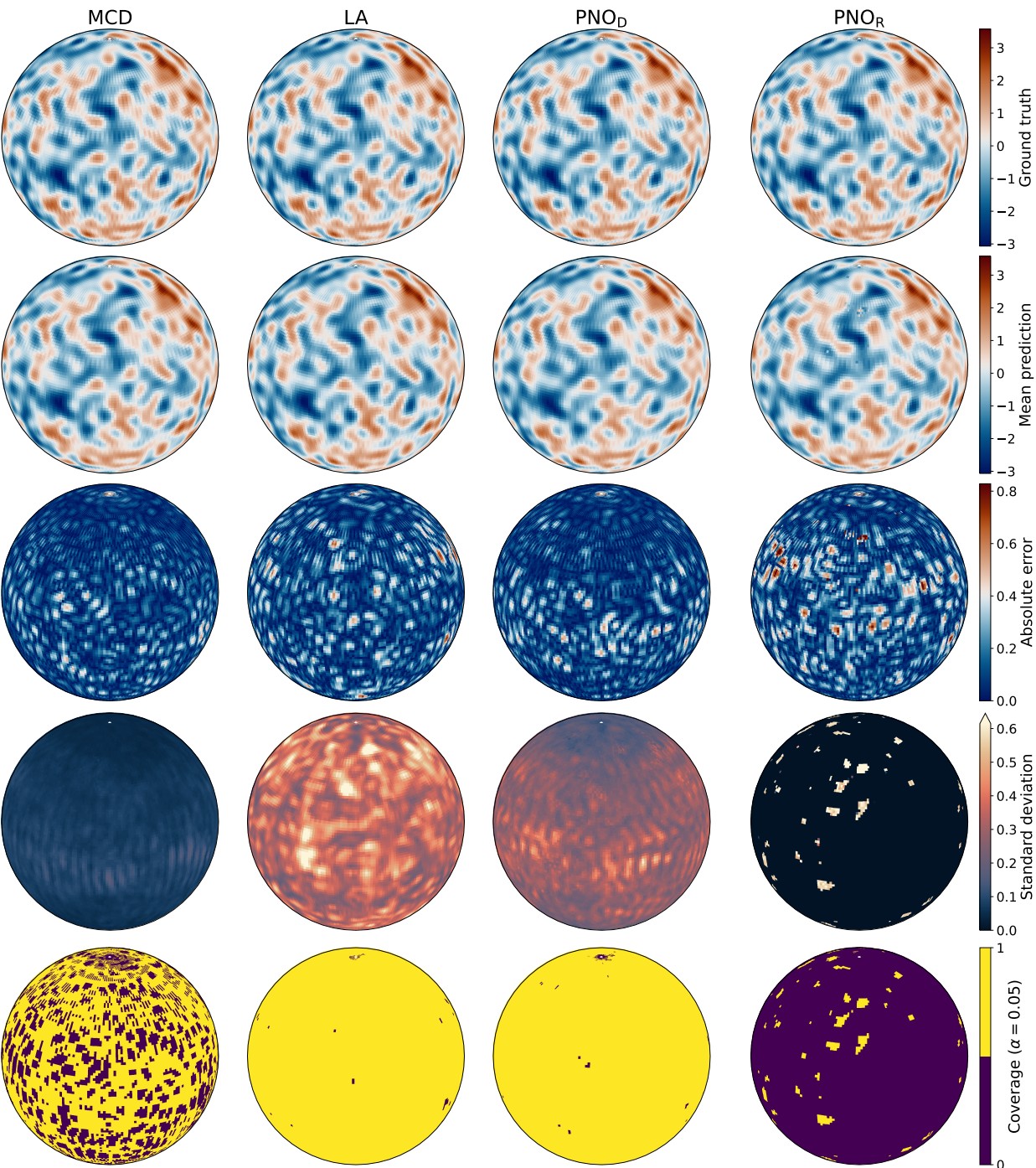

Figure 2: The figure shows the ground truth, mean prediction, absolute error, standard deviation, and coverage for the different methods on a sample of the spherical shallow water equation. The shown variable is the water geopotential and the prediction horizon is one hour with predictions obtained via two-step autoregressive training.

## 4.6 Forecasting of 2m surface temperature

The experiments so far have focused on systems in which the underlying equations are known and computationally tractable. We now want to demonstrate our method on a more complex large-scale dynamical

system with real data. More concretely, we use our approach to create probabilistic medium-range forecasts of the Earth's surface temperature. Recently, neural networks have shown huge success in medium-range weather prediction tasks, with models such as FourCastNet (Kurth et al., 2023), or PanguWeather (Bi et al., 2023). However, while some approaches are available that directly learn probabilistic predictions (Price et al., 2025), most models are either deterministic or require an additional post-processing step (Bülte et al., 2025).

Here, we want to demonstrate the performance of our model in predicting the 2-meter surface temperature across Europe. For computational reasons we restrict the prediction task to only one input variable. We use the ERA5 dataset (Hersbach et al., 2020), provided via the benchmark dataset WeatherBench (Rasp et al., 2024). The data has a spatial resolution of $0.25° \times 0.25°$ and a time resolution of $6h$. For computational reasons, we restrict ourselves to a spatial domain over Europe. We use a 3D spatio-temporal neural operator that takes the last 10 timesteps (60 hours) as the input and predicts the next 10 timesteps. Therefore, we directly predict several future timesteps, instead of using an autoregressive model. The results of the temperature prediction task can be found in Table 4. All models obtain a fairly similar and consistent $L^2$ loss, whereas $PNO_D$ exceeds the alternative methods in the remaining metrics. However, none of the methods obtain an optimal coverage. It is noticeable that the $PNO_R$ leads to a very poor NLL score, while the $PNO_D$ outperforms the other methods significantly with regard to that metric. Figure 3 shows the CRPS and coverage across the spatial domain, aggregated over the test data. The CRPS shows a clear performance distinction between land and sea, which is supported by the findings in Bülte et al. (2025). While for MCDropout and the $PNO_R$, the coverage is clearly different between land and sea, for Laplace and $PNO_D$ this difference is negligible. The latter is preferable, as the models should provide adequate uncertainty even if the underlying spatial domain makes predictions more difficult. Figure 4 shows an additional visualization of the corresponding model predictions.

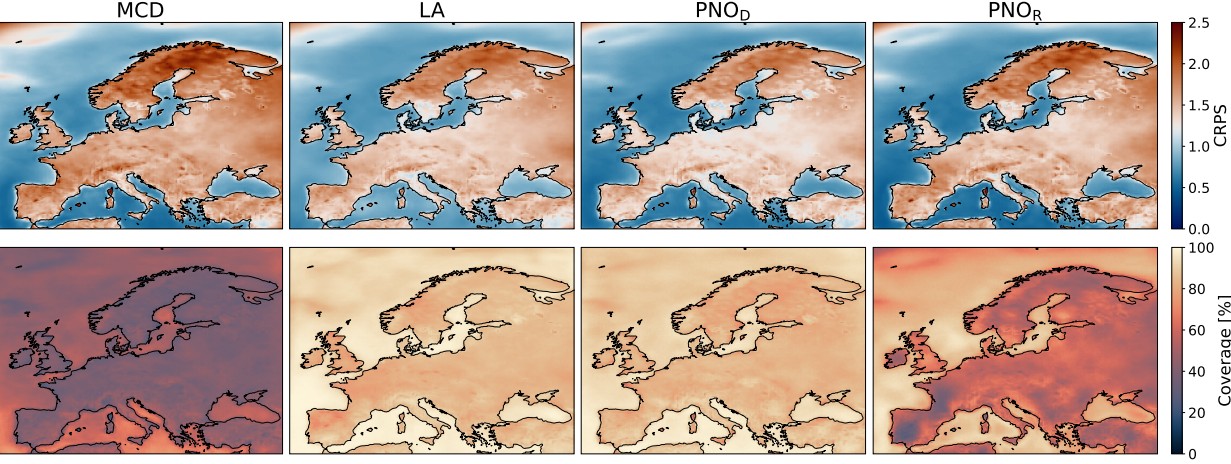

Figure 3: The figure shows the spatial distribution of the CRPS and coverage, aggregated over the test samples for the different methods on the two-meter surface temperature prediction task with a prediction horizon of 24h.

| Model | Method | $\mathcal{L}_{L^2}$ | ES | CRPS | NLL | $\mathcal{C}_{0.05}$ | $|\mathcal{C}_{0.05}|$ |
|---|---|---|---|---|---|---|---|
| **FNO** | DET | 0.0250 (± 0.0001) | 0.0250 (± 0.0001) | 0.0267 (± 0.0001) | - | - | - |
| | $\text{PNO}_\text{D}$ | 0.0242 (±0.0000) | **0.0174** (±0.0000) | **0.0187** (±0.0001) | **-1.9587** (±0.0111) | **0.8623** (±0.0033) | 0.0975 (±0.0012) |
| | $\text{PNO}_\text{R}$ | 0.0241 (±0.0001) | 0.0187 (±0.0001) | 0.0204 (±0.0004) | 616.98 (±1.2×10³) | 0.6536 (±0.0222) | 0.0576 (±0.0022) |
| | MCD | **0.0240** (±0.0001) | 0.0202 (±0.0000) | 0.0214 (±0.0001) | 2.3623 (±0.0237) | 0.4456 (±0.0015) | 0.0351 (±0.0001) |
| | LA | 0.0251 (±0.0001) | 0.0195 (±0.0006) | 0.0210 (±0.0005) | -0.1502 (±1.0099) | 0.6956 (±0.1169) | 0.0802 (±0.0344) |

Table 4: Results for the ERA5 surface temperature prediction task. The best model is highlighted in bold and the standard deviation is given in the brackets.

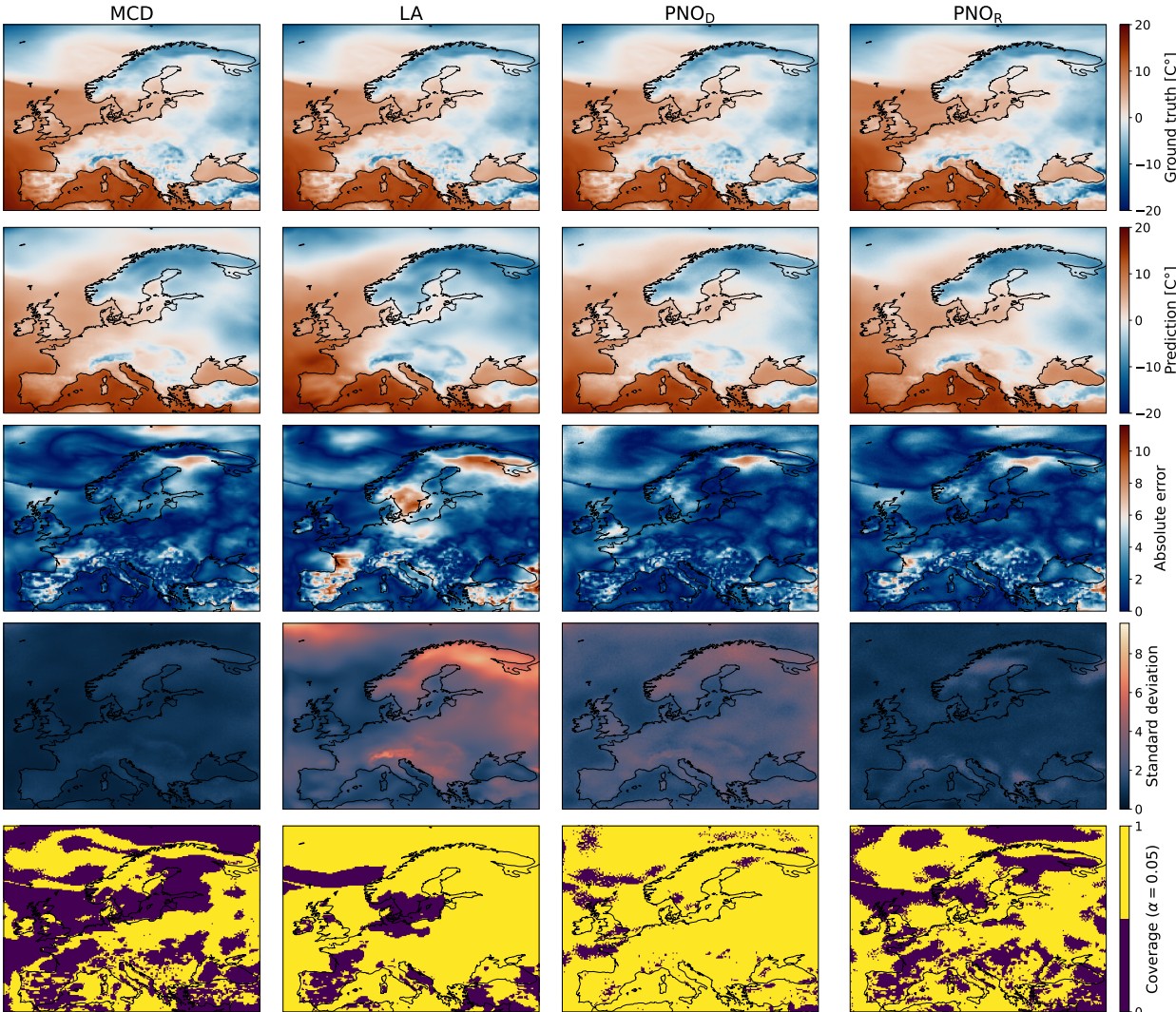

Figure 4: The figure shows the ground truth, mean prediction, absolute error, standard deviation, and coverage for the different methods on a sample of ERA5 dataset. The prediction horizon is 60 hours.

# 5 Discussion

In this work, we introduce PNO, a framework for learning probability distributions over the output function space of neural operators. Our approach combines neural operators with generative modeling utilizing strictly proper scoring rules. We provide a theoretical justification of the approach and analyze how scoring rules can be extended to separable Hilbert spaces. More specifically, we demonstrate how the energy score can be utilized to create a predictive distribution in the output function space of a neural operator. We present and examine different ways of generating a probability distribution using neural operators that are agnostic to the underlying architecture. Our framework enables more accurate uncertainty quantification while maintaining consistent performance in the respective mean prediction task. These advantages are especially noticeable when the underlying system is highly uncertain or chaotic.

We evaluate our approach across several experiments, including stationary PDE solutions, dynamical systems, and weather forecasting. The experiments demonstrate that PNO is transferrable to different network architectures, as well as different domains. Furthermore, PNO performs well in spatio-temporal as well as autoregressive settings. Compared to the baseline methods, PNO shows the best performance across a variety of metrics, especially with regard to the fit between the predictive distribution and the observations. Surprisingly, the mean prediction of the PNO is an improvement against the deterministic model, indicating that modeling the predictive distribution can remove bias in the mean prediction. Finally, the predictions are better calibrated in terms of the coverage of corresponding $1 - \alpha$ prediction intervals.

## 5.1 Limitations

The main limitation of the PNO is the increased computational complexity and memory requirements. These are mainly induced by the energy score, as it requires multiple samples to evaluate. In the case of the $\text{PNO}_\text{D}$ this also requires multiple forward passes of the underlying model. A runtime analysis shows that the training time per epoch of the PNO is around 1.5 - 3 times higher than for the comparing methods. Another limiting factor of PNO is that it inherits the expressivity and (lack of) understanding of the underlying generative model. While $\text{PNO}_\text{R}$ corresponds to a predictive Gaussian in output space, for $\text{PNO}_\text{D}$ the exact distribution and its properties are not available. A better understanding of the predictive distribution could allow for convergence or coverage guarantees. Finally, the results show that PNO leads to unstable predictions for autoregressive inference with long rollout times.

## 5.2 Future work

One of the most important tasks for future research revolves around reducing the computational complexity of our method. As a first step, analyzing the capacities of training a deterministic model and fine-tuning the PNO afterwards appears to be a promising direction. In Appendix E we provide first results in that direction, showing that utilizing fine-tuning can significantly reduce computational costs. In a similar manner, the amount of generated samples could be adapted throughout the training process, increasing performance while reducing training time simultaneously. In addition, other parameters of the training regime could be adjusted to further reduce training and inference time. Another promising research direction is the application of alternative suitable scoring rules such as the Gaussian kernel score. Alternative kernels might induce different characteristics in the predictive distribution and the optimal kernel might be dependent on the underlying data scenario. Future work could also revolve around extending PNO to more complex architectures and larger models. Especially in the context of weather forecasting, models such as the adaptive Fourier neural operator (Kurth et al., 2023) have shown great success and a combination with proper scoring rules might provide a practical method for uncertainty quantification for large-scale systems. In addition, implementing PNO into large-scale models for physical applications such as weather forecasting can lead to improvements in the corresponding uncertainty quantification and extreme event identification, ultimately leading to a deeper understanding of the prediction of such surrogate models.

**Acknowledgements**

C. Bülte and G. Kutyniok acknowledge support by the DAAD programme Konrad Zuse Schools of Excellence in Artificial Intelligence, sponsored by the Federal Ministry of Education and Research.

P. Scholl and G. Kutyniok acknowledges support by the project "Genius Robot" (01IS24083), funded by the Federal Ministry of Education and Research (BMBF), as well as the ONE Munich Strategy Forum (LMU Munich, TU Munich, and the Bavarian Ministery for Science and Art).

G. Kutyniok acknowledges partial support by the Munich Center for Machine Learning (BMBF), as well as the German Research Foundation under Grants DFG-SPP-2298, KU 1446/31-1 and KU 1446/32-1. Furthermore, G. Kutyniok is supported by the GAIn project, which is funded by the Bavarian Ministry of Science and the Arts (StMWK Bayern) and the Saxon Ministry for Science, Culture and Tourism (SMWK Sachsen). G. Kutyniok also acknowledges support from LMUexcellent, funded by the Federal Ministry of Education and Research (BMBF) and the Free State of Bavaria under the Excellence Strategy of the Federal Government and the Länder as well as by the Hightech Agenda Bavaria.

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

# A  Model training and hyperparameters

Across the experiments, all uncertainty quantification methods share the same underlying neural operator architecture. The architectures for the Darcy-Flow experiments are taken from Li et al. (2021) and Rahman et al. (2023) for the FNO and UNO, respectively. An overview of relevant model parameters can be found in table Table 5. As the underlying function space, we use the $L^2$ space and its corresponding norm. The models are trained for a maximum of 1000 epochs and we use early stopping, to stop the training process, if the validation loss does not improve for ten epochs in order to avoid overfitting. For optimization, we employ the popular Adam optimizer (Kingma & Ba, 2017) with gradient clipping. Due to its size, we use an additional learning rate scheduler for the ERA5 dataset, which halves the learning rate if the validation loss does not improve for five epochs. The PNO methods are trained with 3 generated samples, while we use an ensemble size of $M = 100$ across all methods for evaluation. All experiments were implemented in PyTorch and run on an NVIDIA RTX A6000 with 48 GB of RAM. The accompanying code can be found at `https://github.com/cbuelt/pfno`.

| | Model | Batch size | Learning rate | Hidden channels | Projection channels | Lifting channels | Modes |
|---|---|---|---|---|---|---|---|
| Darcy Flow | FNO | 64 | 0.001 | 32 | 256 | 256 | (12,12) |
| | UNO | 64 | 0.001 | [64,128,128, 64,32] | 32 | 32 | [(18,18),(8,8),(8,8), (8,8),(18,18)] |
| KS | FNO | 64 | 0.001 | 20 | 128 | 128 | (10,12) |
| | UNO | 32 | 0.001 | [16,32,64,128, 64,32,16] | 32 | 32 | [(4,20),(4,14),(4,6),(7,6), (7,6),(10,14),(10,20)] |
| SSWE | SFNO | 64 | 0.005 | 32 | 256 | 256 | (32,32) |
| ERA5 | SNO | 16 | 0.005 | 20 | 128 | 128 | (10,12,12) |

Table 5: Overview of relevant model hyperparameters.

# B   Results for the U-shaped neural operator

This section provides additional results for our method using the U-shaped neural operator (Rahman et al., 2023), highlighting the transferability across different architectures. While the model generally performs a bit worse than the FNO on our experiments, the performance of the uncertainty quantification methods shows a similar ordering as before. The $PNO_D$ leads to the best performance on most metrics for Darcy flow and the Kuramoto-Sivashinsky equation. Only once, for the $L^2$ loss for Darcy flow the MCD approach is preferable and $PNO_R$ for the CRPS and the coverage on the KS equation.

| Experiment | Method | $\mathcal{L}_{L^2}$ | ES | CRPS | NLL | $\mathcal{C}_{0.05}$ | $|\mathcal{C}_{0.05}|$ |
|---|---|---|---|---|---|---|---|
| **Darcy Flow** | DET | 0.1949 (± 0.0073) | 0.1949 (± 0.0073) | 0.1565 (± 0.0060) | - | - | - |
| | $PNO_D$ | 0.1894 (±0.0066) | **0.1354** (±0.0050) | **0.1097** (±0.0041) | **-0.2856** (±0.0614) | **0.9069** (±0.0243) | 0.6998 (±0.0467) |
| | $PNO_R$ | 0.2005 (±0.0060) | 0.1473 (±0.0055) | 0.1275 (±0.0044) | 20.018 (±11.926) | 0.5765 (±0.0698) | 0.4218 (±0.0559) |
| | MCD | **0.1875** (±0.0058) | 0.1491 (±0.0076) | 0.1233 (±0.0069) | 2.2782 (±1.4980) | 0.5468 (±0.0600) | 0.2822 (±0.0259) |
| | LA | 0.1954 (±0.0074) | 0.1530 (±0.0052) | 0.1229 (±0.0040) | 0.9996 (±0.2835) | 0.7477 (±0.0411) | 0.4374 (±0.0643) |
| **Kuramoto-Sivashinsky** | DET | 0.9492 (± 0.0250) | 0.9492 (± 0.0250) | 0.7652 (± 0.0241) | - | - | - |
| | $PNO_D$ | **0.9315** (±0.0046) | **0.6560** (±0.0032) | 0.6014 (±0.0088) | **2.0875** (±0.2564) | 0.7828 (±0.0315) | 3.3892 (±0.1214) |
| | $PNO_R$ | 0.9371 (±0.0022) | 0.6595 (±0.0015) | **0.5326** (±0.0023) | 5.4528 (±6.2975) | **0.9110** (±0.0150) | 3.4039 (±0.0442) |
| | MCD | 0.9444 (±0.0014) | 0.8500 (±0.0034) | 0.7163 (±0.0026) | 449.08 (±42.839) | 0.1965 (±0.0050) | 0.4052 (±0.0109) |
| | LA | 0.9494 (±0.0250) | 0.8544 (±0.0265) | 0.6928 (±0.0249) | 54.338 (±37.019) | 0.2560 (±0.0473) | 0.5770 (±0.1251) |

Table 6: Results for the Darcy Flow and Kuramoto-Sivashinsky equation using the UNO architecture. The best model is highlighted in bold for all metrics, except for the interval width, as it is only comparable for models with the same coverage. The standard deviation across the different experiment runs is given in the brackets.

## C  Effect of Fourier dropout

For the PNO$_D$ stochastic forward passes are required to generate the predictive distribution as the output of the neural operator. As mentioned in Subsection 3.2, we use an additional dropout by masking random components in the Fourier domain (Fourier dropout) of each Fourier block in addition to the standard dropout (weight dropout). In the main paper, we always set the same value for the weight dropout as for the Fourier dropout. Here, we provide an additional study to examine the dependence between the different dropout rates and the performance of the PNO$_D$. An FNO architecture is trained for the Kuramoto-Sivashinsky equation with different combinations of the dropout. The corresponding results can be found in Table 7, while Figure 5 shows a corresponding visualization. The dropout rates seem to have a structured effect on the predictions, as the $L^2$ loss and the energy score show better performance with higher dropout rates. However, the NLL seems to be smaller with lower dropout rates. Generally, several of the metrics are minimized by a weight dropout of 0.05 and a higher ($L^2$, ES) or lower (CRPS) Fourier dropout. However, detecting a significant effect is difficult due to the stochastic nature of the approach and the large computational demand for performing detailed experiments. Still, the experiment suggests that different types of dropout seem beneficial, as they induce more stochasticity in the model that could lead to a higher expressivity with regard to the predictive distribution.

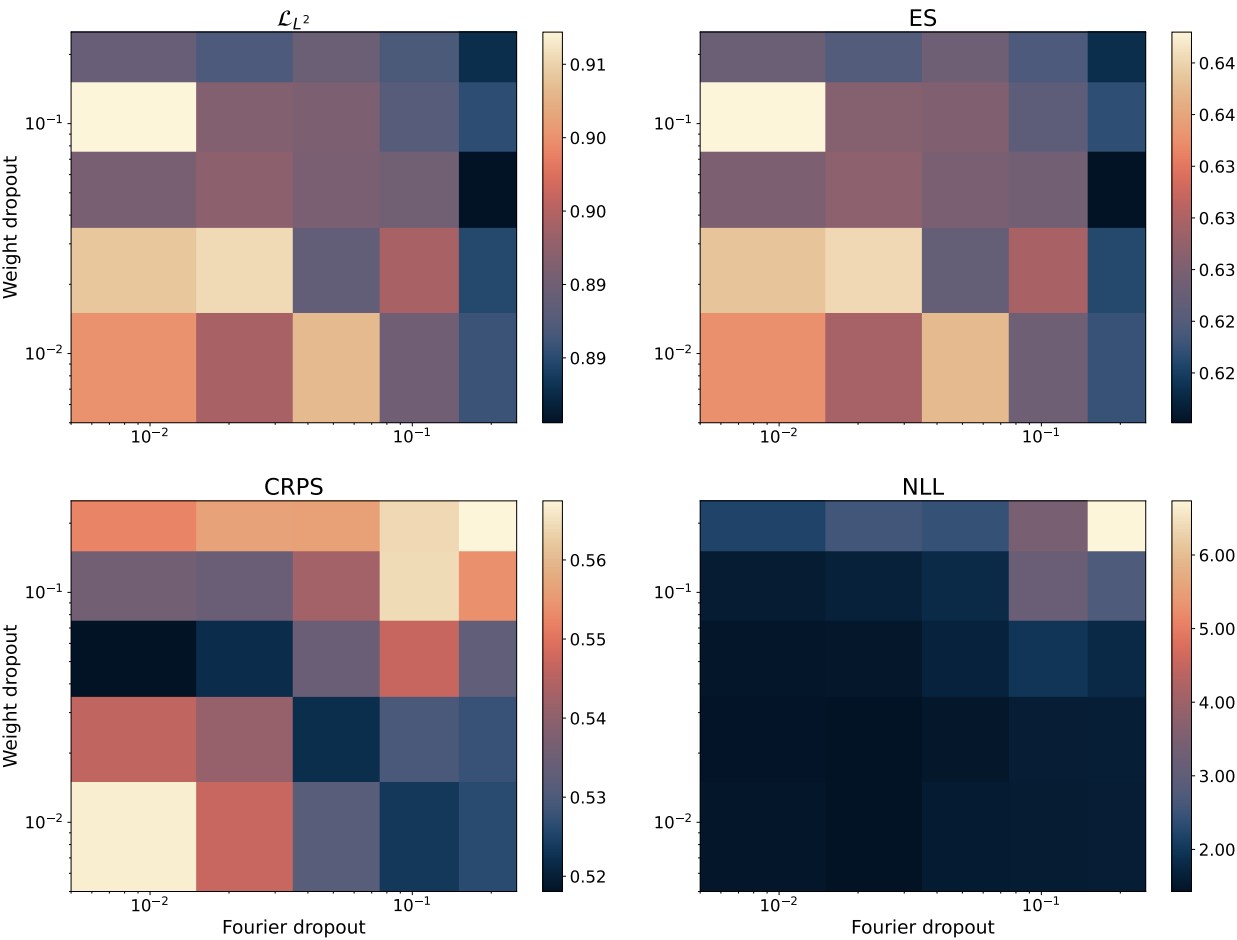

Figure 5: The figure shows the different metrics in dependence of the Fourier and weight dropout for the Kuramoto-Sivashinsky equation. The axes are shown on a log-scale.

| Weight dropout | Fourier dropout | $\mathcal{L}_{L^2}$ | ES | CRPS | NLL | $\mathcal{C}_{0.05}$ | $|\mathcal{C}_{0.05}|$ |
|---|---|---|---|---|---|---|---|
| 0.01 | 0.01 | 0.8998 | 0.6337 | 0.5666 | 1.4866 | 0.8505 | 3.4649 |
| | 0.02 | 0.8942 | 0.6296 | 0.5469 | **1.4281** | 0.8746 | 3.4825 |
| | 0.05 | 0.9032 | 0.6361 | 0.5315 | 1.5714 | 0.8366 | 3.2629 |
| | 0.1 | 0.8899 | 0.6266 | 0.5236 | 1.5835 | 0.8336 | 3.2151 |
| | 0.2 | 0.8858 | 0.6237 | 0.5263 | 1.6129 | 0.8294 | 3.2427 |
| 0.02 | 0.01 | 0.9042 | 0.6367 | 0.5462 | 1.4577 | 0.8983 | 3.6815 |
| | 0.02 | 0.9054 | 0.6376 | 0.5410 | 1.4299 | 0.9051 | 3.6425 |
| | 0.05 | 0.8887 | 0.6259 | 0.5223 | 1.5018 | 0.8667 | 3.3519 |
| | 0.1 | 0.8942 | 0.6298 | 0.5296 | 1.6132 | 0.8294 | 3.2363 |
| | 0.2 | 0.8848 | 0.6231 | 0.5276 | 1.6191 | 0.8279 | 3.2578 |
| 0.05 | 0.01 | 0.8907 | 0.6274 | **0.5181** | 1.4798 | 0.8712 | 3.2865 |
| | 0.02 | 0.8922 | 0.6284 | 0.5221 | 1.4968 | 0.8660 | 3.3190 |
| | 0.05 | 0.8908 | 0.6273 | 0.5342 | 1.6776 | 0.8198 | 3.2360 |
| | 0.1 | 0.8900 | 0.6268 | 0.5466 | 1.9742 | 0.7766 | 3.1370 |
| | 0.2 | **0.8806** | **0.6201** | 0.5327 | 1.7949 | 0.8078 | 3.2122 |
| 0.1 | 0.01 | 0.9072 | 0.6390 | 0.5357 | 1.5733 | 0.8418 | 3.2286 |
| | 0.02 | 0.8914 | 0.6280 | 0.5344 | 1.6708 | 0.8151 | 3.1668 |
| | 0.05 | 0.8910 | 0.6276 | 0.5425 | 1.8126 | 0.7929 | 3.1337 |
| | 0.1 | 0.8877 | 0.6255 | 0.5643 | 3.1770 | 0.7391 | 3.0874 |
| | 0.2 | 0.8852 | 0.6234 | 0.5537 | 2.7249 | 0.7638 | 3.0821 |
| 0.2 | 0.01 | 0.8891 | 0.6264 | 0.5520 | 2.2083 | 0.7598 | 2.9729 |
| | 0.02 | 0.8870 | 0.6249 | 0.5563 | 2.5560 | 0.7431 | 2.9557 |
| | 0.05 | 0.8895 | 0.6266 | 0.5561 | 2.4287 | 0.7527 | 3.0111 |
| | 0.1 | 0.8869 | 0.6247 | 0.5640 | 3.4221 | 0.7352 | 2.9904 |
| | 0.2 | 0.8828 | 0.6217 | 0.5675 | 6.7351 | 0.7319 | 3.0469 |

Table 7: Metrics for different combinations of dropout for training on the Kuramoto-Sivashinsky equation.

## D    Runtime analysis

Since using a scoring rule as the loss function requires predicting multiple samples for each optimization step, a runtime analysis of PNO is of high interest. In Table 8, we show the estimated runtime of MCD, LA, $\text{PNO}_D$, and $\text{PNO}_R$. As expected, both PNO methods require significantly more time per epoch, with $\text{PNO}_D$ taking about three times as long as the baseline methods and $\text{PNO}_R$ about 50% longer. The main bottleneck is the evaluation of the energy score, which requires to calculate a distance matrix for all predictive samples. Furthermore, $\text{PNO}_D$ requires several forward passes per optimization step, increasing the runtime even further. $\text{PNO}_R$ on the other hand requires more memory and had to be trained with a smaller batch size (16 instead of 32), which also increases the computing time per epoch compared to MCD and LA, which are both trained without uncertainty.

| Method | Avg. Number of Epochs | Time per Epoch (s) | Total Time (s) |
|--------|----------------------|--------------------|----------------|
| MCD | **112.89** | 22.36 | **22718** |
| LA | 140.00 | **21.56** | 27161 |
| $\text{PNO}_D$ | 132.33 | 58.71 | 69920 |
| $\text{PNO}_R$ | 209.78 | 31.92 | 60272 |

Table 8: Runtime comparison on SSWE with one-step autoregressive training. The number of epochs is averaged across the ten training runs using early stopping with 10 epochs of patience. The timings are averaged over 25 training epochs on an RTXA6000 and the best models are highlighted in bold.

# E   Improving computational efficiency

As previously mentioned, the PNO requires substantially more memory and computing time than the deterministic neural operator. In this section, we briefly want to present a strategy to mitigate this behavior. The most straightforward way to improve computing time is to consider a given pre-trained (deterministic) model, which is a realistic scenario in a practical setting. In that case, the $PNO_D$ can be used to fine-tune that model, as it only requires stochastic dropout with nonzero probability. Table 9 compares the runtime and metrics for the deterministic model, the $PNO_D$ and the fine-tuned $PNO_D$. The results show that the pre-trained model requires substantially fewer epochs and training time, while at the same time leading to an improved performance. Figure 6 shows an additional visualization of the training and validation loss against the training time of the model. While training a deterministic model and a $PNO_D$ together, does not yield significant improvements in the computing time, the pre-trained $PNO_D$ requires significantly fewer epochs to converge and therefore can be efficiently used if one has an already trained deterministic model.

|  | $\mathbf{PNO_D}$ | $\mathbf{DET}$ | $\mathbf{PNO_D}$ **(pre-trained)** |
|---|---|---|---|
| $t$ | 7043.38 ($\pm$ 895.79) | 4689.33 ($\pm$ 408.54) | **2404.10** ($\pm$ 700.00) |
| $\varnothing$ epochs | 150.78 ($\pm$ 19.18) | 112.33 ($\pm$ 9.79) | **46.33** ($\pm$ 13.49) |
| $\mathcal{L}_{L^2}$ | 0.8793 ($\pm$ 0.0072) | **0.8635** ($\pm$ 0.0048) | 0.8702 ($\pm$ 0.0046) |
| ES | 0.6195 ($\pm$ 0.0049) | 0.7541 ($\pm$ 0.0049) | **0.6130** ($\pm$ 0.0033) |
| CRPS | 0.5496 ($\pm$ 0.0097) | 0.5974 ($\pm$ 0.0058) | **0.5343** ($\pm$ 0.0080) |
| NLL | **3.0389** ($\pm$ 0.5872) | 107.1038 ($\pm$ 22.9412) | 6.0401 ($\pm$ 1.9573) |
| $\mathcal{C}_{0.05}$ | 0.7640 ($\pm$ 0.0191) | 0.3600 ($\pm$ 0.0082) | **0.7950** ($\pm$ 0.0154) |
| $|\mathcal{C}_{0.05}|$ | 3.0852 ($\pm$ 0.0422) | 0.6046 ($\pm$ 0.0164) | 3.1327 ($\pm$ 0.0288) |

Table 9: Performance comparison of the pre-trained $PNO_D$ against the deterministic model and the regular $PNO_D$ using ten different model runs. The standard deviation across the runs is given in the brackets and the best model is highlighted in bold.

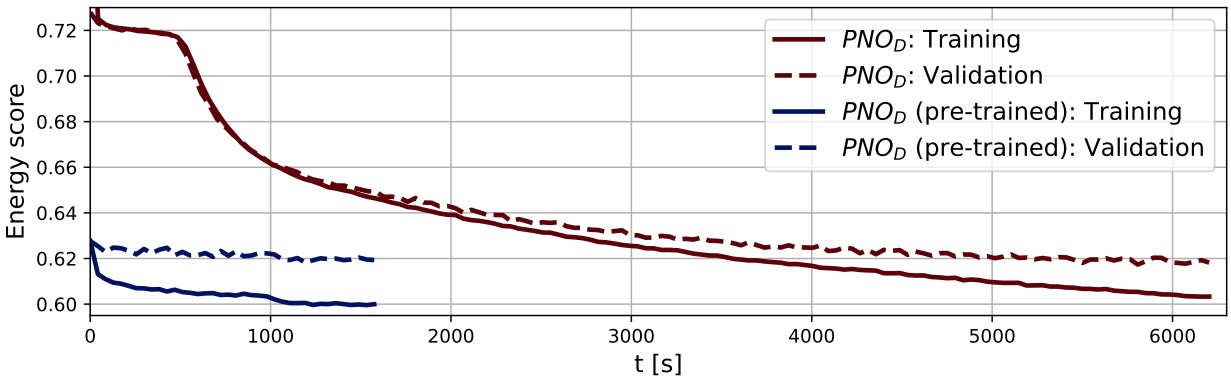

Figure 6: The figure shows the training time against the energy score for the regular $PNO_D$ and the pre-trained $PNO_D$ for a specific experiment run. The solid and dashed lines show the training and validation loss, respectively.

# F Influence of the number of predictive samples on the performance of PNO

Our PNO approach depends on utilizing an unbiased estimator of the energy score in Equation 4, which requires the model to generate $M > 1$ samples. While in practice we found that even for only $M = 3$ generated samples, the model shows improved performance, increasing the sample size is mainly hindered by computational reasons. In order to further analyze the model training in dependence of the sample size $M$, we trained both PNO approaches on the Kuramoto-Sivashinsky equation with different training sample sizes $M$. For evaluation, all models still use an ensemble size of one hundred members across the test data. Table 10 shows the corresponding results, which are again aggregated over ten runs. For the $PNO_R$, the $L^2$ loss and energy score already perform well for $M = 3$ samples, while the other metrics are improved by increasing the sample size. While the training time per epoch increases, the differences seem negligible for less than twenty samples. On the contrary, the $PNO_D$ shows the best performance, if $M \geq 10$ samples are already used during the training process. Especially the negative log-likelihood improves significantly. However, for the $PNO_D$, the training time per epoch increases drastically, as the model also requires $M$ stochastic forward passes. On the other hand, for the $PNO_D$, the number of training epochs decreases with the sample size. A solution might be using different training regimes, starting with a low sample size and increasing it during the training process, to reduce the computational demand.

| Model | $M$ | $t$ | ∅ epochs | $\mathcal{L}_{L^2}$ | ES | CRPS | NLL | $\mathcal{C}_{0.05}$ |
|---|---|---|---|---|---|---|---|---|
| **$PNO_D$** | 3 | **37.72s** | 151 | 0.8793 (±0.0072) | 0.6195 (±0.0049) | 0.5496 (±0.0097) | 3.0389 (±0.5872) | 0.7640 (±0.0191) |
| | 5 | 44.58s | 154 | 0.8790 (±0.0059) | 0.6191 (±0.0038) | 0.5577 (±0.0080) | 5.2872 (±1.8802) | 0.7541 (±0.0189) |
| | 10 | 59.94s | 126 | 0.8799 (±0.0021) | 0.6196 (±0.0015) | **0.5348** (±0.0038) | 2.3471 (±0.2235) | 0.8001 (±0.0095) |
| | 20 | 90.64s | 135 | **0.8784** (±0.0091) | **0.6188** (±0.0061) | 0.5566 (±0.0088) | 5.3147 (±1.5163) | 0.7592 (±0.0170) |
| | 50 | 181.45s | **98** | 0.8857 (±0.0050) | 0.6239 (±0.0036) | 0.5356 (±0.0030) | **1.9174** (±0.1097) | **0.8036** (±0.0092) |
| **$PNO_R$** | 3 | 31.08s | 163 | **0.8640** (±0.0038) | **0.6081** (±0.0027) | 0.4743 (±0.0037) | 1.2268 (±0.0083) | 0.9401 (±0.0030) |
| | 5 | **29.75s** | 146 | 0.8661 (±0.0036) | 0.6096 (±0.0025) | 0.4751 (±0.0032) | 1.3768 (±0.4510) | 0.9421 (±0.0027) |
| | 10 | 31.35s | 152 | 0.8649 (±0.0036) | 0.6088 (±0.0025) | 0.4737 (±0.0038) | 1.2180 (±0.0158) | 0.9437 (±0.0020) |
| | 20 | 35.06s | **135** | 0.8680 (±0.0053) | 0.6110 (±0.0037) | 0.4756 (±0.0051) | 1.2198 (±0.0181) | **0.9453** (±0.0027) |
| | 50 | 43.43s | 181 | 0.8649 (±0.0035) | 0.6088 (±0.0025) | **0.4733** (±0.0027) | **1.2138** (±0.0085) | 0.9449 (±0.0028) |

Table 10: Comparison of the performance of $PNO_D$ and $PNO_R$ in dependence of the training sample size $M$ for the Kuramoto-Sivashinsky equation. The best results are highlighted in bold, where $t$ denotes the average training time per epoch.

# G   Comparison of PNO$_D$ and PNO$_R$

In this section we compare the two presented approaches PNO$_D$ and PNO$_R$ in more detail and provide some guidance as to which method might be more useful for a specific problem set. While section 4 evaluates both approaches with respect to different metrics and experiments, these absolute values make it difficult to see which method is generally preferable. For that purpose we here also report the skill scores across different metrics, which are a common tool for assessing relative improvements over a reference model. The skill score for a performance metric over a test set is given by

$$\text{SS}_F = \frac{\overline{\text{S}}_{\text{ref}} - \overline{\text{S}}_F}{\text{S}_{\text{opt}} - \overline{\text{S}}_{\text{ref}}},$$

where $\overline{\text{S}}_F$ denotes the average performance of the method of interest, $\overline{\text{S}}_{\text{ref}}$ denotes the corresponding average metric of a reference method and $\text{S}_{\text{opt}}$ denotes the optimal achievable value. For the $L^2$, ES and CRPS metrics, $\text{S}_{\text{opt}}$ corresponds to zero. For the other metrics, the skill score is not applicable (for the negative log-likelihood $\text{S}_{\text{opt}}$ would be minus infinity). The skill score is positively oriented with a maximum of one, negative values indicating worse performance than the reference, and zero indicating no improvement. Table 11 shows the skill scores of all methods with respect to the deterministic baseline. The results show that in the cases where PNO$_D$ outperforms PNO$_R$, the relative improvement is quite high, while in the opposite direction, PNO$_R$ only achieves a relatively small improvement against PNO$_D$. Averaged over all experiments, it is evident that PNO$_D$ is preferable to all metrics, as it generally leads to an improvement against PNO$_R$.

| Experiment | $\mathcal{L}_{L^2}$ | ES | CRPS |
|---|---|---|---|
| Darcy Flow | 0.2509 | 0.303 | 0.3471 |
| KS | -0.0178 | -0.0188 | -0.1588 |
| SSWE | 0.2328 | 0.191 | 0.4216 |
| ERA | -0.0059 | 0.0704 | 0.0836 |
| ∅ | 0.115 | 0.1364 | 0.1734 |

Table 11: Skill scores for the different experiments and PNO$_D$ with respect to PNO$_R$ as a baseline. The last row shows the average over all experiments.

While from a performance perspective, PNO$_D$ is preferable, there might be other considerations determining which model to choose. If one has a fixed architecture or an already trained model, PNO$_D$ might be a better choice, as it is easier to incorporate into a given model, requiring only a nonzero dropout probability, while PNO$_R$ also requires additional weights. On the other hand, PNO$_D$ comes with a higher computational cost, thus if the computational resources are limited, PNO$_R$ might be preferable. Finally, one might choose the method based on knowledge of the underlying data distribution and the corresponding expressivity of the method. Both methods, PNO$_R$ and PNO$_D$ only offer a crude approximation to the posterior predictive distribution (Hron et al., 2018). While for PNO$_D$ it is in general very difficult to make any statements about the expressivity and approximation quality of the resulting distribution, for the PNO$_R$ case, the resulting distribution is a standard Gaussian. Therefore, if the data distribution is known to be similar to a Gaussian, PNO$_R$ might be a suitable method. On the other hand, if the data distribution is known to be complex or multimodal, PNO$_D$ might perform better, although no guarantees can be made that it matches the underlying distribution.

# H    Additional figures

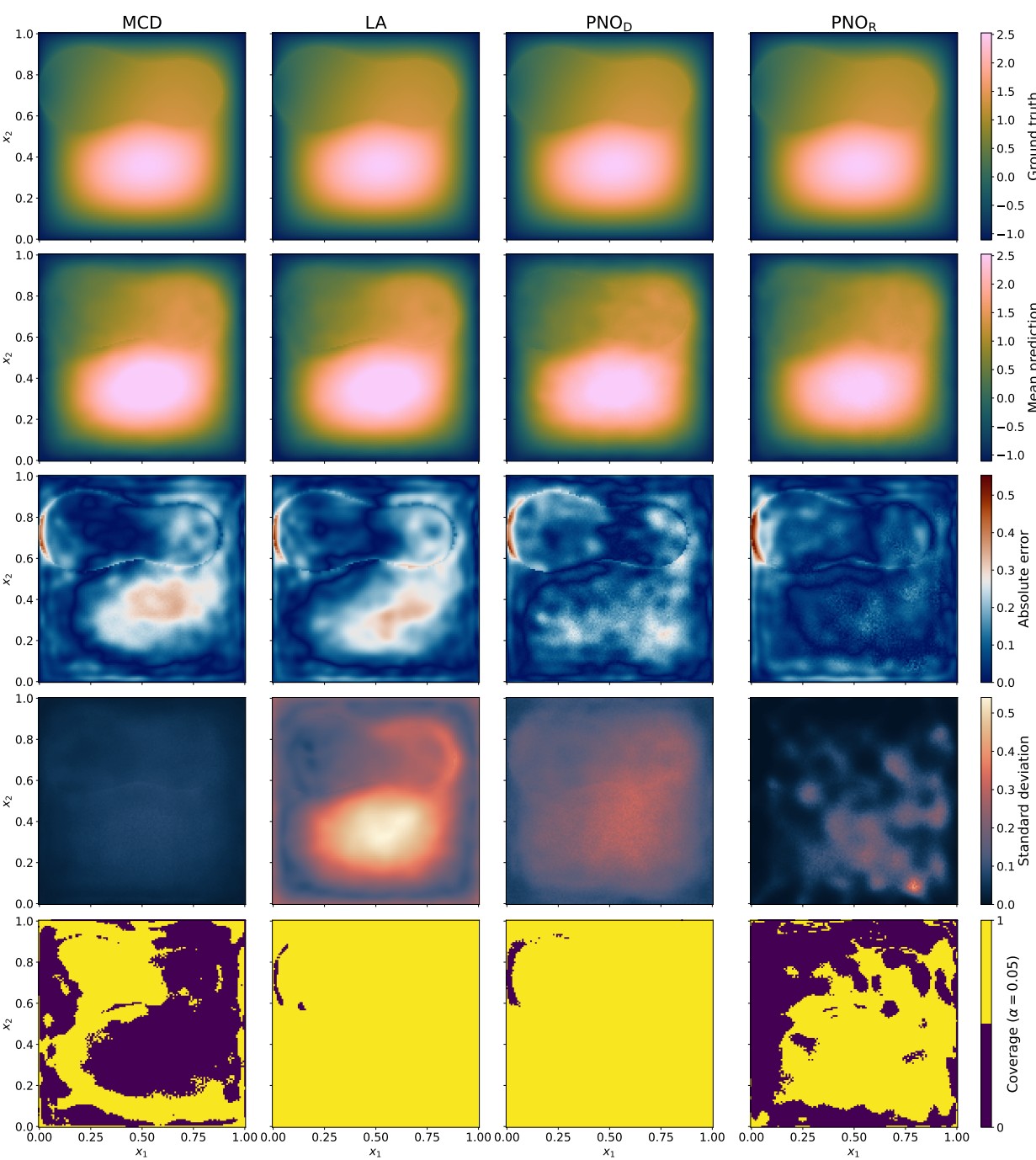

Figure 7: The figure shows the ground truth, mean prediction, absolute error, standard deviation, and coverage for the different methods on a sample of the Darcy-Flow equation.

# I   Additional results for SSWE

| $\Delta t$ | Method | $\mathcal{L}_{L^2}$ | ES | CRPS | NLL | $\mathcal{C}_{0.05}$ | $|\mathcal{C}_{0.05}|$ |
|---|---|---|---|---|---|---|---|
| **1h** | DET | 0.3215 (± 0.0106) | 0.3215 (± 0.0106) | 0.1260 (± 0.0035) | - | - | - |
| | PNO$_D$ | **0.2955** ($\pm$0.0207) | **0.2202** ($\pm$0.0132) | **0.0896** ($\pm$0.0051) | **-0.4609** ($\pm$0.0463) | **0.9840** ($\pm$0.0032) | 0.8923 ($\pm$0.0467) |
| | PNO$_R$ | 0.3132 ($\pm$0.0244) | 0.2208 ($\pm$0.0170) | 0.1048 ($\pm$0.0069) | $2.5\times10^7$ ($\pm3.6\times10^7$) | 0.6912 ($\pm$0.0897) | 0.4971 ($\pm$0.0842) |
| | MCD | 0.3845 ($\pm$0.0206) | 0.2972 ($\pm$0.0194) | 0.1193 ($\pm$0.0102) | 0.4516 ($\pm$0.5340) | 0.7063 ($\pm$0.0496) | 0.3819 ($\pm$0.0123) |
| | LA | 0.4222 ($\pm$0.0511) | 0.3063 ($\pm$0.0345) | 0.1346 ($\pm$0.0135) | 0.1821 ($\pm$0.2472) | 0.9024 ($\pm$0.0694) | 0.9606 ($\pm$0.2178) |
| **10h** | DET | **1.1346** (± 0.1861) | **1.1346** (± 0.1861) | **0.4771** (± 0.0665) | - | - | - |
| | PNO$_D$ | $2.8\times10^3$ ($\pm8.3\times10^3$) | $2.7\times10^3$ ($\pm8.0\times10^3$) | 22.3235 ($\pm$65.581) | **6.0553** ($\pm$0.7769) | 0.4695 ($\pm$0.0207) | 43.754 ($\pm$128.49) |
| | PNO$_R$ | $9.4\times10^3$ ($\pm2.0\times10^4$) | $7.6\times10^3$ ($\pm1.6\times10^4$) | $3.6\times10^3$ ($\pm7.7\times10^3$) | $1.7\times10^{20}$ ($\pm4.6\times10^{20}$) | 0.3056 ($\pm$0.1644) | $1.3\times10^4$ ($\pm3.2\times10^4$) |
| | MCD | 1.3324 ($\pm$0.2282) | 1.2066 ($\pm$0.2177) | 0.5043 ($\pm$0.0739) | 25.394 ($\pm$3.3598) | 0.2511 ($\pm$0.0047) | 0.4104 ($\pm$0.0304) |
| | LA | 1.5445 ($\pm$0.2696) | 1.2835 ($\pm$0.2493) | 0.5412 ($\pm$0.1041) | 7.2994 ($\pm$6.0597) | **0.5254** ($\pm$0.1033) | 1.0291 ($\pm$0.2616) |

Table 12: Results for the spherical shallow water equation with single-step training. The best model is highlighted in bold and the standard deviation is given in the brackets.

