# OpenReview forum: "Probabilistic neural operators for functional uncertainty quantification"
_TMLR — Accepted by TMLR_

### Review · Reviewer_AXTt · 2025-02-11

**Summary Of Contributions:**

Their key contribution is to propose the use of energy score, a widely used scoring rule, as the target to have neural operators learn the distribution of possible solutions. Their approach maps both the coefficient field and a hidden noise to a possible solution. They present two variants - PNO-D using stochastic dropout and PNO-R using the reparameterization trick. The work is thoroughly validated across multiple datasets. While one of the variants will perform well on a given dataset, neither variant has consistently superior performance across all test cases, leaving room for improvement in developing more robust approaches.

**Audience:**

Yes

**Broader Impact Concerns:**

None that I think need to be specifically highlighted here.

**Claims And Evidence:**

Yes

**Requested Changes:**

The authors should provide a more detailed analysis of when and why each variant performs better, as well as rule-of-thumb guidelines for choosing between variants based on problem type.

**Strengths And Weaknesses:**

Strengths:
- The paper introduces energy scores as a principled approach to learn solution distributions, supported by the proof that the energy score is a strictly proper scoring rule in a separable Hilbert space.
- The authors validate their approach across multiple PDE problems and real-world data, which show both its capabilities and limitations quantitatively.

Weaknesses:
- The method introduces significant computational costs. The loss function of PNO-D is especially slow to compute.
- Neither variant demonstrates consistently superior performance across all test cases, and there's no clear guide on which variant to choose for a specific problem.

---

> ### Author Response · Authors · 2025-02-21
>
> We would like to thank reviewer AXTt for the feedback and mentioned improvements regarding our work.  We believe that the reviewer has raised some important points regarding our work and, following the reviewer's suggestion, addressed the raised issues and uploaded a new version of the pdf:
>
> **The method introduces significant computational costs. The loss function of PNO-D is especially slow to compute.**
>
> We agree that this is the main bottleneck of our approach. In order to address that issue, we provide an additional analysis (Appendix E), where we show how fine-tuning an already existing model can greatly reduce computational cost, while keeping improved performance. This seems an adequate scenario in application settings, where one might already have a trained deterministic model, but later decides to analyze uncertainties. In that case, $PNO_D$ converges very quickly and requires significantly less computing time than it does in training from scratch.
>
> **The authors should provide a more detailed analysis of when and why each variant performs better, as well as rule-of-thumb guidelines for choosing between variants based on problem type.**
>
> We agree with reviewer AXTt that from the experimental section it is not clear to identify a method with superior performance. However, if looking at relative improvement to a baseline, it becomes evident that the $PNO_D$ is overall preferable. We included an additional section (Appendix G), where we compare the skill score of the predictions (i.e. relative improvement against a baseline). While there exist cases where $PNO_R$ performs better than $PNO_R$, in these cases the improvement is only marginal, while the other way around $PNO_D$ achieves significant improvement against $PNO_R$. This shows that in general, $PNO_D$ is preferable to $PNO_R$, which is hard to see from the absolute evaluation in section 4 alone. In addition to the aforementioned analysis, we also provided some more guidance and rule-of-thumb selection criteria for the corresponding methods in Appendix G.

---

### Review · Reviewer_fAH7 · 2025-02-12

**Summary Of Contributions:**

The paper introduces a novel framework that extends neural operators by learning the probability distributions over function spaces instead of single deterministic predictions.
It further provides uncertainty quantification for PDEs by incorporating this uncertainty directly into the training process.
They prove that the energy score (scoring rule) remains valid in separable Hilbert spaces, which ensures the theoretical side of their approach.
The framework has flexible sampling strategies using dropout or reparameterization tricks to make it compatible with several neural operator architectures in a plug-and-play manner.
Extensive experiments on several PDE problems and weather forecasting have provided robust results of PNO compared to traditional uncertainty quantification methods.

**Audience:**

Yes

**Claims And Evidence:**

Yes

**Requested Changes:**

Please consider below suggestions, which are not critical but can help to improve the work.
- Discuss more regarding the *excessive computational overhead* due to the multiple forward passes along with *potential strategies/attempts* to address this issue. For example, considering the framework’s plug‐and‐play nature, can you provide results on how integrating architectures like the Adaptive Fourier Neural Operator help to reduce some of this overhead burden or solve the scalability issue?
- Please clarify/elaborate the *reliance on specific dropout or Gaussian assumptions* and discuss how these can affect the flexibility in learning  *more complex uncertainty patterns*.

**Strengths And Weaknesses:**

**Strengths**
- Proves that the energy score is strictly proper in separable Hilbert spaces.
- Transforms neural operators from deterministic solvers into models that learn long-term probability distributions over function spaces by incorporating uncertainty directly during training.
- Supports multiple sampling strategies and various neural operator architectures.
- Shows better results on several PDE problems and real world scenario- weather forecasting.

**Weaknesses**
- Requires multiple forward passes or samples per training step (increased memory and runtime).
- The increased computations can limit applicability to very large-scale or high-resolution problems.
- Can lead to instability or growing error in long-term autoregressive forecasting scenarios.
- The reliance on specific dropout or Gaussian assumptions may potentially still restrict flexibility in capturing complex uncertainty patterns.

---

> ### Author Response · Authors · 2025-02-21
>
> We would like to thank reviewer fAH7 for the feedback and mentioned improvements regarding our work. We believe that the reviewer has raised some important points regarding our work and, following the reviewer's suggestion, addressed the raised issues and uploaded a new version of the pdf.
>
> **Increased computational costs**
>
> We agree that the computational cost is a main issue of our method and potential attempts to improve the runtime therefore quite important. In addition to the runtime analysis, we now provide a way to significantly reduce the training time, by utilizing a pre-trained deterministic model. This is also realistic in application settings, where a user might already have a pre-trained model but requires additional uncertainty estimates. Before we only briefly mentioned this in the future work section of the conclusion, but now we provide detailed experiments on the KS equation in Appendix E.
>
> **Can lead to instability or growing error in long-term autoregressive forecasting scenarios**
>
> It is true that the PNO can lead to instabilities in long-term autoregressive forecasting scenarios. However, the same goes for the competing methods, as can be seen in Section 4.5 and Table 3. When using a two-step autoregressive training regime (as suggested in [1]), PNO outperforms the other methods. While the instabilities are still noticeable (for example in the NLL of $PNO_R$), this comes down to the general problem of combining probabilistic predictions with autoregressive forecasts. It is not clear, what is the best way to obtain stable uncertainty estimates for each time step, while still training the model in an autoregressive manner.
>
> **Please clarify/elaborate the reliance on specific dropout or Gaussian assumptions and discuss how these can affect the flexibility in learning more complex uncertainty patterns.**
>
> We agree that the reliance on specific dropout or Gaussian assumptions may potentially be restricting. Both methods might be rather crude approximations of an arbitrary probability distribution. However, the same holds for almost any approximations such as the Laplace approximation, since large models, such as neural operators, are heavily overparametrized. In order to keep methods feasible a certain amount of restrictions is needed. However, if moving to more complex settings to generate probability distributions, our methodology would remain the same. Training via the energy score could potentially be combined with any method that generates a probability distribution, as long as it allows for gradient-based optimization. In order to further elaborate on the difference between dropout and Gaussian reparametrization, we introduced an additional section in the appendix (Appendix G), where we compare both methods and also provide a broad overview of assumptions and expressivity.
>
> [1] Bonev et. al. Spherical Fourier Neural Operators: Learning Stable Dynamics on the Sphere, 2023.

---

### Review · Reviewer_irfP · 2025-02-15

**Summary Of Contributions:**

This paper proposes a probabilistic neural operator, which extend the neural operator frameworks into probabilistic (mostly forecasting based) settings, producing distributions over operators (or functions). The main application is in forecasting partial differential equations or physical systems, where the function is over the entirety of the domain of the system (e.g. the region that we wish to predict the weather). The authors give several ways of training the probabilistic neural operator by constructing an energy scoring rule and finding a way to approximate its gradient using samples from a generative base model.

The paper builds on Pacchiardi et al, ’24 who come up with a method for differentiating through a proper scoring rule (not necessarily a log likelihood) by using a generative model. In the Bayesian setting, the generative model is the posterior predictive distribution. Then, the generative model is compared to either the real observation or an adversary.

The proposed energy score is a specific type of kernel, and then Thm 3.1 follows standard kernel scoring rule analyses (Steiwart & Ziegel, ’21). For finite posterior samples, Thm 3.1 reduces to Eq. 7, which is the sample based approximation used to train the network. Eq. 7 requires posterior predictive samples to evaluate the scoring rule and the kernel aspect, which is achieved by either dropout or (what is essentially) a variational approximation.

Experiments are performed on a couple of simulated differential equations, where the proposed method tends to do reasonably well. It seems like neither version strongly outperforms the other, which is a shame. Perhaps a bit more work on the dropout method or on the parameterization of the variational distribution would help.

**Audience:**

Yes

**Claims And Evidence:**

Yes

**Requested Changes:**

-	PNO_R and PNO_D are not defined. It took me quite a while to realize that PNO_D is the dropout version of the score (eq 7), while PNO_R is another variant without posterior samples by using reparameterization. What does Eq 7 simplify to in the PNO_R setting?

-	Sec 4.2 – I would probably state (again?) that `u` is the given heldout observation, while P_\theta^M is the probability distribution over functions that we are predicting. It would be good practice to go through and define each of these metrics explicitly in terms of each variable used in them (and then these equations can go in the appendix to save space). As of now, it’s a bit unclear what \bar u_M – is it the posterior predictive mean or is it averaged over all `u`?

-	Additionally, use \bar \sigma^2 instead of \hat \sigma^2 to keep the naming the same for variables that use the same thing.

-	Table 1: a bit more editorializing here and in the following tables would be good for understanding if you’re going to leave these as a table (and not say a bar plot). For example, “PNO_D tends to perform best across both architectues, with the FNO architecture tending to have better performance [due to the inductive bias of the problem].” Otherwise, as a reader, I’m not sure what the bold in the lower table is supposed to show / represent and and why PNO_D is not bolded for coverage in the UNO side of the table.

-	“if the correct distribution is learned, then C_\alpha(x) = 1-\alpha… “ The following statement here is a bit imprecise. Ideally, we would want the coverage to be exactly $1 - \alpha$, with under-calibration occurring (e.g. overfitting) if the estimated region covers a smaller region than 1-\alpha, and over-calibration occurring if the estimated region covers a larger region than 1 - \alpha. Thus, we’d probably want the C_0.05 band to be “closest” to 0.95, and the width of the prediction interval (the next column) to be as small as possible.

-	Sec 4.5: “PNO_R leads to extremely large values for the negative log likelihood …” Indeed, this could be quite problematic for decision making and forecasting. Often, this is due to improper parameterization of \sigma in the reparameterization trick. How was \sigma parameterized here?

-	Sec 4.5: “MCD approach does not lead to stable predictions for the ten hour…” This is probably because for forecasting with dropout, you need to modify how the dropout is done by dropping out the same units and weights at each time step in the rollout to get consistently dropped out paths like in Gal & Ghahramani, neurips, 16. We can also see this visually in the MCD version of Figure  2 where the coverage failure is uniform, as well as the standard deviations for both dropout models there as well. By comparison, LA is at least somewhat varying between regions (as is PNO_R).

-	Sec 3.2: Dropout is an extremely crude posterior predictive approximation that isn’t really theoretically grounded (see Hron, Mathews, & Ghahramani, “variational Bayesian dropout: pitfalls and fixes, ICML, ’18), and really neither is the explicit gaussian reparameterization form that you’re using from Kingma et al, ’15 due to the lack of the likelihood term. Of course, many Laplace approximations are also similarly quite crude for neural networks due to the overparameterization of the model.

However, as the authors correctly point out, using dropout does introduce noise into the training process, which might be reasonable for producing semi reasonable uncertainty estimates. It could be better to explicitly parameterize a Gaussian posterior over the weights and toss the scoring rule in a stochastic variational inference objective (Hoffman et al, JMLR, ’13) with corrections for the generalization from the likelihood itself into the scoring rule (Lacoste-Julien et al, AISTATS, ’11 and Maus et al, Neurips, ’24 come to mind).

The takeaway here is that the claim of “we wish to take into account uncertainty during training” is not sufficient a priori to justify PNO_D over PNO_R or the other Laplace approximations. Thus, I would suggest toning down those claims.

-	Section 4.2: “if the correct distribution is learned, then C_\alpha(x) = 1-\alpha..”, this isn’t strictly true as we could have extremely underconfident predictive distributions that have C_\alpha >> 1- \alpha (as is seen in some of the experimental results). I’d suggest rephrasing this sentence to something like “if the correct distribution is learned, then C_\alpha(x) is close to 1-\alpha”.

-	Section 4.3: “PNO_R leads to extremely large values for the negative log likelihood..” This probably suggests a failure in optimization (e.g. the posterior variance collapsed to zero), which is interesting here because in many situations with Bayesian NNs there’s severe underfitting.


References:

Pacchiardi et al, Probabilistic forecasting with generative networks via scoring rule minimization, JMLR, 2024.

Lacoste-Julien et al, Approximate inference for the loss calibrated Bayesian, AISTATS, 2011.

Maus et al, Approximation aware Bayesian optimization, Neurips, 2024.

Hron, Mathews, & Ghahramani, “variational Bayesian dropout: pitfalls and fixes, ICML, ’18

Hoffman et al, Stochastic variational inference, JMLR, 2013.

Gal & Ghahramani, A Theoretically Grounded Application of Dropout in Recurrent Neural Networks, Neurips, 2016.

**Strengths And Weaknesses:**

Strengths:

-	The paper is mostly well written and the technical details are clearly described.

-	The paper nicely applies pre-existing techniques (variational posterior approximations and kernel scoring rules) into a very useful application.

-	Experiments look like they’re evaluated on problems of real world interest.

Weaknesses:
-	Missing comparisons to cited relevant work such as Guo et al, ’23, and Magnani et al, ’24, Ma et al, ’24.

-	Personally, I’d also strongly appreciate the “basic” baselines of deep ensembles (simply retraining 3x models), a single model (CRPS score becomes MAE) and a heteroscedastic predictor (predict mean and variance for the predictor).

-   see also the comments below.

---

> ### Author Response · Authors · 2025-02-21
> **Main comments**
>
> We would like to thank Reviewer irfP for the detailed feedback and the very helpful insights.  We believe that the reviewer has raised some important points regarding our work and, following the reviewer's suggestion, addressed the raised issues and uploaded a new version of the pdf:
>
> **Missing comparisons to cited relevant work such as Guo et al, ’23, and Magnani et al, ’24, Ma et al, ’24.**
>
> We implement the Laplace approximation, which was used by [1,2]. The main difference to Magnani et. al. '24 is that we do not incorporate the additional Gaussian currying part, for which no code is provided. We feel that by utilizing the Laplace approximation in the function space of neural operators, we capture the main ideas of [1,2]. For completeness, we added the corresponding reference to Magnani et. al. '24 also in Section 4.1.\
> The methodology of Guo et. al. [3] is specifically designed for DeepONets, which are also a class of neural operators but substantially differ from the Fourier neural operator and its variants, which we utilize in this paper. While their approach uses interesting ideas for uncertainty quantification, incorporating it into the FNO architecture would require additional theoretical work. Furthermore, the authors did not provide code for their method.\
> Finally, regarding Ma et. al. [4] we did not implement the corresponding method as it is not directly comparable in our evaluation setting. Their method is based on training an additional neural operator that predicts error bounds for the deterministic prediction, utilizing conformal prediction. However, this does not provide a posterior predictive distribution and therefore does not allow for evaluation using most of our metrics. While the deterministic model could be evaluated with the L2-loss, energy score, and CRPS, these will be the same as the deterministic baseline, which we now added (see next comment). While the results on the coverage would be interesting, we feel that it still does not fit that well into our evaluation setting. In principle, we would expect the method to obtain a better coverage, as it is optimized for that exact task via the conformalized prediction regime. However, apart from the coverage, no comparison could be made.
>
>
> **Personally, I’d also strongly appreciate the “basic” baselines of deep ensembles (simply retraining 3x models), a single model (CRPS score becomes MAE) and a heteroscedastic predictor (predict mean and variance for the predictor).**
>
> We agree that the "basic" baselines can offer great insights and grounds for comparison. Therefore, we included a deterministic model as an additional baseline, for which the energy score reduces to the L2 loss and the CRPS to the mean absolute error. We decided to choose that baseline, as in an application scenario, one might already have a pre-trained model and might be interested in the improvements offered by the PNO approach. The new results show that the PNO (and the other methods mostly) perform better than the deterministic baseline in the mean prediction. This shows that learning a predictive distribution can improve performance even in the mean prediction.\
> Regarding deep ensembles, the main bottleneck is the computational time, as a single model can already take several days to train (with results aggregated over ten different seeds). Training an ensemble of $M=100$ then becomes infeasible. While in principle we could include an ensemble with only three members, the difference will be negligible to the deterministic model, as the low sample size does not allow for good estimates of the variance or the coverage.\
> While we think that a heteroscedastic predictor would also be a good idea, it might be difficult to formulate it exactly in the function space setting, which then would be with regard to a (functional) Gaussian process. While we could of course just formulate the predictor in a discretized setting, we feel that this would work a bit against the main notion of neural operators and this paper, where we derive the underlying theory in the corresponding function spaces.\
> By including the deterministic baseline we feel that we fill a possible gap in the evaluation process and obtain an evaluation process that is thorough enough to justify our model. However, we are of course interested in your thoughts on this. Please also note that since we are still running the updated autoregressive models (compare paragraph 6.), the deterministic baseline result for the SSWE task is still missing. We expect to update this final metric in the next week.
>
> **It seems like neither version strongly outperforms the other**
>
> This is an interesting aspect, that we investigate further in our updated version. We provide now an additional analysis of relative performance in Appendix G, which shows that $PNO_D$ is preferable in general. See also our answer to reviewer AXTt for details. We thank the reviewer for the insightful points and are open to any additional feedback.

---

> > ### Comment · Reviewer_irfP · 2025-03-09
> > **Thanks**
> >
> > Thanks for the clarifications and updates to the paper, I think that it reads much more clean now.
> >
> > Unfortunately, your math rendering in the comment below didn't quite resolve so I'm a bit confused there - but i think i have some understanding of the point.
> >
> > >(with results aggregated over ten different seeds). ...  While in principle we could include an ensemble with only three members,
> >
> > In my mind, this is enough for 3 ensembles of 3 seeds each which gives an estimate of model variance as well, and the coverage may not be too bad.

---

> ### Author Response · Authors · 2025-02-21
> **Minor comments 1/2**
>
> We thank the reviewer for the insightful points and are of course open to any additional feedback and ideas. In addition to the previous points, we also address the reviewer's suggestions regarding more minor details.
>
> **PNO_R and PNO_D are not defined.**
>
> Thank you for pointing this out! We added a definition of $PNO_D$ and $PNO_R$ in Section 3.2.
>
> **What does Eq 7 simplify to in the PNO_R setting?**
>
> Although the Gaussian case seems more tractable than the dropout case, Eq 7, does not admit a closed-form analytical expression for $PNO_R$. As shown in [5], even for a standard multivariate normal distribution, the first term of the energy score has no analytical solution, requiring us to sample from the corresponding Gaussian and to calculate the energy score via Eq 7.
>
> **Clarify the definition of the metrics and move them to the Appendix.**
>
> We agree that the definition of the metrics was not exactly clear and updated the corresponding Section 4.2. We added (and revisited) some further notation at the beginning of that section. While it might take up some space, we believe that it is advantageous to show the full expressions of the evaluation metrics in the main paper, as otherwise, the corresponding definitions might be unclear in the experimental section.
>
> **Additionally, use \bar \sigma^2 instead of \hat \sigma^2 to keep the naming the same for variables that use the same thing.**
>
> While it might seem inconsistent, we think that it might be beneficial to stick with the notation of $\bar{u}_M$ and $\hat{\sigma}^2$, since $\bar{u}_M = \frac{1}{M}\sum^M\_{m=1} u^m$ is the actual average of $P\_\theta^M\$, while $\hat{\sigma}^2$ and $\hat{u}\_M$ are estimators of $\mathbb{V}[P\_\theta^M]$ and $u$, respectively. This notation seems more in alignment with classical statistical notation.
>
> **Editorializing the tables.**
>
> We agree that the tables and the corresponding descriptions might be difficult to understand and can be improved. First, in accordance with changing the description of $\mathcal{C}_\alpha$ (see next paragraph), we now highlight the coverage in bold which is closest to $0.95$. The interval width is not put in bold, as it is only comparable if several methods achieve a similar coverage, which is generally not the case in our experiments. We included a more thorough description in the description of Table 1.\
> Furthermore, since we added an additional baseline, which will make the tables larger and take up more space in the main part of the paper, we decided to move all results regarding the UNO architecture to the appendix (Appendix B). This makes Section 4 more concise and, in our opinion, the paper more digestible and visually appealing. However, this could of course be reverted, if the reviewers disagree.
>
> **“if the correct distribution is learned, then $C_\alpha(x) = 1-\alpha$… “ The following statement here is a bit imprecise.**
>
> We agree that the statement regarding the coverage is imprecise, as we only focused on over-calibration. We adjusted the corresponding part of the paper, now stating that in the ideal case, an estimator should obtain $\mathcal{C}_{\alpha} \approx 1- \alpha$ while keeping the interval width as small as possible.
>
> **Sec 4.5: “PNO_R leads to extremely large values for the negative log likelihood …” Indeed, this could be quite problematic for decision making and forecasting. Often, this is due to improper parameterization of \sigma in the reparameterization trick. How was \sigma parameterized here?**
>
> We parametrized $\sigma(x)$ using the projection layer of the FNO with a softplus activation function and an added threshold of $1e-6$ to improve numerical stability. This parametrization is the same across all experiments, for which reason we believe that the unusually high values of the NLL might be due to the specific experiment and its autoregressive nature. We agree that the high values suggest a collapse of the variance, especially since the other metrics are still similar to the competing methods. We adjusted the corresponding description of the $PNO_R$ in Section 4.
>
> **Sec 4.5: “MCD approach does not lead to stable predictions for the ten hour…” This is probably because for forecasting with dropout, you need to modify how the dropout is done by dropping out the same units and weights at each time step in the rollout to get consistently dropped out paths like in Gal & Ghahramani, neurips, 16.**
>
> We thank the reviewer for the insightful comment. We agree that this might be a good way to circumvent the coverage failure. Therefore, we implement a new version of the autoregressive model, where the dropout is consistent across the autoregressive paths, but varies across the different samples in order to create a probability distribution. While we already implemented the new version, the experiments for the SSWE dataset are still running and we expect to udpate the corresponding results within the next week.

---

> ### Author Response · Authors · 2025-02-21
> **Minor comments 2/2**
>
> **Dropout is an extremely crude posterior predictive approximation that isn’t really theoretically grounded [...] and really neither is the explicit gaussian reparameterization form that you’re using [and] many Laplace approximations are similarly quite crude... The takeaway here is that the claim of “we wish to take into account uncertainty during training” is not sufficient a priori to justify PNO_D over PNO_R or the other Laplace approximations. Thus, I would suggest toning down those claims.**
>
> We thank the reviewer for the insightful comments and for providing similar works in the direction of stochastic variational inference and scoring rules. We agree that both, the dropout, as well as the Gaussian approximation are rather crude and might potentially be not expressive enough to approximate any arbitrary probability distribution. However, as the reviewer mentioned, this is also the case for other approximations. Providing analytically tractable full posteriors for models such as the Fourier neural operator is extremely difficult and choosing some useful approximation might be the only way to require uncertainty estimates or probabilistic predictions. However, we agree that so far, we might have chosen a rather strong wording, which seems to indicate that "incorporating uncertainty into training" implies that the $PNO$ is preferable against other architectures. We rather want to point to the fact that this approach has been rarely applied in the literature, with more work focusing on post-hoc methods. We adjusted the wording in the corresponding sections, as suggested by the reviewer.
>
> [1] Weber et. al. Uncertainty quantification for
> fourier neural operators, 2024\
> [2] Magnani et. al. Linearization Turns Neural Operators into Function-Valued Gaussian Processes, 2024\
> [3] Guo et. al. IB-UQ: Information bottleneck based uncertainty quantification for neural function regression and neural operator learning, 2023\
> [4] Ma et. al. Calibrated Uncertainty Quantification for
> Operator Learning via Conformal Prediction, 2024\
> [5] Szekely, Gabor. E-statistics: Energy of statistical samples, 2013

---

> ### Author Response · Authors · 2025-03-03
> **Implementation of consistent dropout paths**
>
> **Sec 4.5: “MCD approach does not lead to stable predictions for the ten hour…” This is probably because, for forecasting with dropout, you need to modify how the dropout is done by dropping out the same units and weights at each time step in the rollout to get consistently dropped out paths like in Gal & Ghahramani, neurips, 16.**
>
> As mentioned in the previous comments, we now updated the deterministic baseline also for the SSWE autoregressive task. In addition, we followed the ideas of the reviewer and experimented with an autoregressive version of the models that works by dropping out the same units and weights and each time step. For clarification, our original approach was a version where each timestep is predicted in a deterministic fashion (either by a deterministic prediction for MCD/Laplace or a mean prediction for PNO), and uncertainty is only estimated on the final timestep of interest. While we implemented a new version with consistent dropout paths, where in each timestep different samples are drawn and propagated through the autoregressive predictions, the results did not show improvement against the previous approach.
> For that reason, we decided to stick with the original version, where the uncertainty is only considered in the final step. Although Figure 2 definitely shows that there are some issues with the dropout approaches, as mentioned by the reviewer, these already appear in the one hour prediction and are therefore not necessarily an artifact of the autoregressive rollout. In general, using a neural operator with an intrinsic time domain, as in the KS or ERA5 examples, seems to work much more stable for all involved methods. Since the autoregressive performance is not the main focus of the paper, we believe that the results are sufficient as they are now and leave further investigation of combining PNO with autoregressive steps for future work.
>
> We again thank the reviewer for the helpful feedback and ideas.

---

> ### Author Response · Authors · 2025-03-11
>
> Thank you for the update and the further comments. We apologize for the poorly rendered math; we have fixed it now.
>
> Regarding the deep ensembles. We use the ten seeds only for evaluation in order to achieve robust results. So, 3 ensembles of  3 seeds would only give us 3 predictions that we can use to compute the coverage. Since we are computing 95% confidence intervals, an ensemble of $M=3$ predictions is far from ideal to compute the coverage. However, we did a quick experiment on the Kuramoto-Sivashinsky equation regarding your suggested setup with 3 ensemble members, averaged across three seeds, as can be seen in the following table:
>
> | Model  | L2 | ES | CRPS | NLL | Coverage | Interval width |
> |-----------|---------|---------|---------|---------|---------|---------|
> | Ensemble  | 0.8849  | 0.7256  | 0.5865  | 414.1504 | 0.2106  | 0.3785  |
> | PNO-D  | 0.8793  | 0.6195  | 0.5496  | 3.0389  | 0.7640  | 3.0852  |
> | PNO-R  | 0.8640  | 0.6081  | 0.4743  | 1.2268  | 0.9401  | 3.2594  |
>
>
> Compared to the other methods in the paper, the L2 loss, ES, and CRPS are worse than the PNO methods and similar to the other baselines. However, the NLL and Coverage are quite poor for the ensemble, as expected when using only $M=3$ ensemble members, whereas the comparing methods use $M=100$.
> While these results show that deep ensembles can provide a good baseline (and might perform well with a higher ensemble size), we want to stress that this method is computationally extremely demanding, and providing a proper ensemble of $M=100$ for the different experiments is practically not feasible. On the other hand, as we want to provide a robust evaluation, using different seeds for training is crucial to assess the performance metrics and their standard deviation.
> Therefore, we believe that it fits the paper better to not include the deep ensemble as an additional metric, as we would need to break up the seeds, therefore making the different methods not directly comparable.
>
> We would like to thank you again for the feedback and the interesting ideas regarding our paper.

---

### Decision · Action_Editor_4ECc · 2025-03-24

**Recommendation:** Accept with minor revision

**Comment:**

The paper studies an important problem of probabilistic neural operators. The reviewers kindly provided insightful suggestions that the authors incorporate in the current draft.

Throughout the discussion, it came out that despite MCDropout was first extended to neural operators in Ma et al. (2024) "Calibrated Uncertainty Quantification for Operator Learning via Conformal Prediction" it was not clear in the current writing of this paper. The authors are encouraged to integrate this edit.

**Audience:**

Yes, the "ML on functions" community is interested in this work.

**Claims And Evidence:**

The paper's claims are supported by theoretical justification, and their assumptions are validated, to some extent, by empirical studies.

---

> ### Author Response · Authors · 2025-03-27
>
> We thank the area chair for the positive feedback and for accepting our paper. We included the suggested reference in the related work section (last paragraph of 1.2), and in the baselines section (first paragraph in 4.1).